# Copper catalysis at operando conditions—bridging the gap between single nanoparticle probing and catalyst-bed-averaging

David Albinsson [1], Astrid Boje [1], Sara Nilsson[1], Christopher Tiburski [1], Anders Hellman[1,2], Henrik Ström [3] & Christoph Langhammer [1✉]

In catalysis, nanoparticles enable chemical transformations and their structural and chemical fingerprints control activity. To develop understanding of such fingerprints, methods studying catalysts at realistic conditions have proven instrumental. Normally, these methods either probe the catalyst bed with low spatial resolution, thereby averaging out single particle characteristics, or probe an extremely small fraction only, thereby effectively ignoring most of the catalyst. Here, we bridge the gap between these two extremes by introducing highly multiplexed single particle plasmonic nanoimaging of model catalyst beds comprising 1000 nanoparticles, which are integrated in a nanoreactor platform that enables online mass spectroscopy activity measurements. Using the example of CO oxidation over Cu, we reveal how highly local spatial variations in catalyst state dynamics are responsible for contradicting information about catalyst active phase found in the literature, and identify that both surface and bulk oxidation state of a Cu nanoparticle catalyst dynamically mediate its activity.

[1] Department of Physics, Chalmers University of Technology, 412 96 Göteborg, Sweden. [2] Competence Centre for Catalysis, Chalmers University of Technology, 412 96 Göteborg, Sweden. [3] Department of Mechanics and Maritime Sciences, Chalmers University of Technology, 412 96 Göteborg, Sweden. ✉email: clangham@chalmers.se

The performance of a heterogeneous catalyst is determined by how the catalyst material interacts with its environment[1]. Therefore, it is one of the key objectives of catalysis science in general, and of single particle catalysis in particular, to establish so-called structure-function relationships where the state of the catalyst, down to the level of the individual active nanoparticle, is directly correlated with formed products[2–4]. Historically, studies to create these structure-function correlations have relied on methods that use (ultra) low pressure, which is far from the conditions at which the catalysts are intended to operate. This is problematic because it, for example, has been shown that the surface or bulk state of a catalyst can be strongly affected by the pressure of the surrounding gas environment, leading to what is commonly referred to as the pressure gap[5]. Therefore, the development of operando experimental techniques, with the aim to determine the state of the active catalyst under application-relevant conditions, has been a major tour de force in catalysis during the last decades[6–8]. However, although such operando techniques today can provide fascinating information about the catalyst material and its state down to the atomic level, they typically treat the whole catalyst bed as a single entity in a single state when correlating activity with structure. This, despite only probing structure or state in a very small region of the bed, or even only on a single nanoparticle. In other words, the catalyst activity is measured after the entire catalyst bed and thus integrated over all of the catalyst material in the system, irrespective of its specific local state, which may vary significantly across the bed due to temperature and reactant concentration gradients. Consequently, this combination of integral activity measurements with single point probing of catalyst material state is prone to produce erroneous structure-function relationships[9,10]. In attempts to alleviate this problem, recent studies have used a combination of techniques to both spatially and temporally investigate model[3,11–17] and more complex catalysts[9,11,12,18–22], with several comprehensive reviews on the topic published recently[2,9,10,23]. For example, the thin 150 μm capillary used in Spaci-MS[2] enables spatiotemporal characterization of reactant gradients with a spatial resolution of ca. 300 μm inside catalyst monoliths. However, due to the size of the capillary, it is not possible to directly measure concentration gradients inside the porous catalyst material itself, since the pores typically are more than 1000 times smaller. Moreover, none of the previous approaches offers single nanoparticle resolution with respect to operando detection of catalyst state, while at the same time being able to probe the entire catalyst bed.

In response, we introduce highly multiplexed operando single particle plasmonic nanoimaging in combination with our recently developed nanofluidic reactor platform[24,25] that enables online mass spectrometry from extremely small amounts of catalyst material. As the key advance, we demonstrate how this concept is able to combine single particle resolution in catalyst state dynamics imaging from a quasi-2D array of 1000 nominally identical Cu nanoparticles that constitute the whole catalyst bed, with online quadrupole mass spectrometry (QMS) from the same particles. This combination of simultaneous averaged local information with multiplexed single particle resolution from the exact same sample uniquely resolves how reactor geometry and specific location in the catalyst bed induce widely different reaction conditions for the individual nanoparticles.

To showcase and analyze these effects, we design two types of nanofluidic reactors with distinctly different properties and we study carbon monoxide (CO) oxidation over a Cu model catalyst. CO oxidation is one of the most studied reactions in heterogeneous catalysis, largely due to its applications in pollution mitigation and its relevance in fundamental research[26]. In these contexts, noble metals (e.g., Pt, Pd, Rh, and Ru) have proven to perform well and are thus widely used. However, due to their scarcity and high cost, there is a growing interest in alternative catalyst materials, with Cu as one attractive candidate. However, despite extensive research on Cu-based CO oxidation catalysts, there is still no consensus in the literature about which oxidation state of the Cu catalyst is the most active[15,25–29]. This situation can be explained by the difficulty in maintaining a single oxidation state of Cu under relevant experimental conditions in a catalyst bed, as pointed out by Jernigan et al., who in an early work, identified a decrease in activity as the extent of Cu oxidation increased[30]. This has more recently been confirmed by some ambient pressure studies[15,28]. However, another study, utilizing ambient pressure XPS, identified $Cu_2O$ as the most active state[29], which has also been proposed previously[26]. Even more recently, using environmental TEM, it has been suggested that the surface oxide structure dynamically changes during reaction[31]. This persistent disagreement thus calls for further experimental investigations using operando concepts, and Cu is, therefore, the system of our choice for the present study.

We have divided the paper into two stages, where the first analysis level focuses on the effects of reactant concentration gradients, and the second one on the individual particle responses of the 1000 nanoparticles present in the catalyst bed. Thereby, we reveal that the activity dynamics of the individual particles generally depend on reactor geometry, that they are drastically different even within catalyst bed sub-sections (patches) containing 100 nanoparticles each, and that they are not resolved by the patch-averaged response in either reactor. This is notably despite the fact that also the patch-averaged response itself reveals activity dynamics that depend on the patch position in the catalyst bed. These findings shed light on how highly local activity and state dynamics of individual catalyst nanoparticles may be responsible for often contradicting information about catalyst active phase found in the literature in general, and for the studied Cu catalyst system during the oxidation of CO by oxygen ($O_2$) in particular[15,26–29].

## Results

**Reactor chip and model catalyst bed design.** We have micro-/nanofabricated miniaturized mimics of two types of catalytic reactors into Si/$SiO_2$ wafers, using the same principles that we have recently introduced[25], and as described in detail in the "Methods" section, Supplementary Methods and Supplementary Fig. 2. Specifically, we have designed two types of reactor chips, where the reaction chamber either maximizes or minimizes the formation of reactant concentration gradients across the model catalyst bed, i.e., a first reactor that is well-mixed and a second reactor that is of the plug-flow type. In both designs, the bed contains 1000 nominally identical nanofabricated Cu nanoparticles of 120 nm diameter and 40 nm height, which are arranged in 10 regular array patches containing 100 nanoparticles each placed 20 μm apart, as detailed in Fig. 1. In each of these patches, the individual Cu nanoparticles are spaced at a pitch of 900 nm. This provides a reasonably high catalyst particle packing density, while at the same time ensuring particle-particle distances larger than the diffraction limit of visible light. In this way, it becomes possible to resolve changes in light scattering intensity from the individual Cu nanoparticles in each array, as further explained below.

Each chip consists of a microfluidic system (Fig. 1a) that handles gas flow to and from a nanofluidic reaction zone where the model catalyst bed is located (Fig. 1b). We use the term nanofluidic here due to the out-of-plane-dimension of our system, which is only 100 nm. The chip itself is mounted in a sample holder acting as an interface for gas and electrical

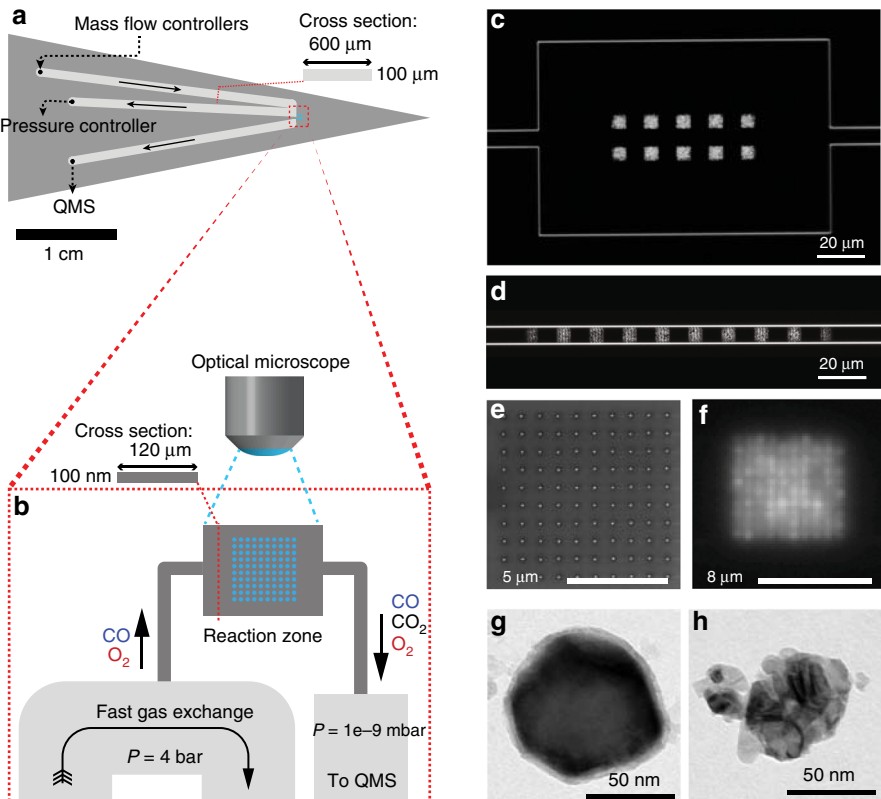

**Fig. 1 Reactor chip layout and optical readout. a** Schematic of the micro- and nanofabricated reactor chip comprising a microfluidic in- and outlet system that connects to the model catalyst bed, as well as the high-pressure gas handling system and the QMS, respectively. The cross-sectional dimensions of the microfluidic channels are also shown. Arrows indicate flow direction. **b** Schematic depiction of the model catalyst bed (reaction zone) of the chip that contains the catalyst nanoparticles and the corresponding cross-sectional dimensions of the well-mixed reactor in (**c**). Note that the schematics are not drawn to scale. The entire catalyst bed is imaged by dark-field optical microscopy that resolves all particles individually based on their optical scattering signature. **c** Dark-field scattering image of a well-mixed catalyst bed with dimensions 180 μm × 120 μm × 100 nm (L × W × H), containing ten Cu nanoparticle array patches, each comprising 100 individual particles with a diameter of 120 nm and a thickness of 40 nm. **d** Dark-field scattering image of the plug-flow catalyst bed consisting of a straight channel with 10 μm × 100 nm cross section and decorated with ten array patches, each comprising 100 individual particles with the same dimensions as in (**c**). **e** SEM image of a 100 Cu nanoparticle array patch. **f** Zoom in dark-field image of a 100 Cu nanoparticle array patch. **g** TEM image of a representative single Cu nanoparticle reduced in 1.2% $H_2$ at 400 °C. The slight surface oxide stems from exposure to air during transport to the TEM. **h** TEM image of a representative Cu nanoparticle oxidized in 0.2% $O_2$ at 400 °C, revealing the dramatic structural changes oxidation can induce.

connections (Supplementary Fig. 3). This design enables online QMS measurements of the Cu catalyst bed with an estimated active surface area of 26 $\mu m^2$ in the present study (Details in Supplementary Methods). For the well-mixed-type reactor we fabricated a reaction zone with dimensions 180 μm × 120 μm × 100 nm (L × W × H), and connected it to a 10 μm wide channel on the out- and inlet side (Fig. 1c). For the plug-flow-type model catalyst bed, we fabricated a straight channel reaction zone with 10 μm × 100 nm cross-sectional dimensions and a total reaction zone length of 216 μm (Fig. 1d). These designs result in reactant residence times of approximately 6 ms and 60 ms in the plug-flow-type and well-mixed reactor, respectively. According to unified flow model calculations[32] the pressure-driven flow through the reactors results in a pressure drop of 0.05 bar (2.42–2.37 bar, Supplementary Fig. 4) across the reaction zone and finite-volume simulations reveal close to identical local $O_2$ concentration for each patch for the well-mixed-type (<7% variation, Fig. 2a, c). In contrast, the plug-flow reactor exhibits a 0.59 bar pressure drop (2.73–2.14 bar) and significantly different local $O_2$ concentration between the first and the last patch (72% variation, Fig. 2b, c, Supplementary Fig. 4 and Supplementary Methods for details).

The Cu nanoparticle positions in these two bed types are defined by electron-beam lithography and grown by electron

beam evaporation of Cu through the nanofabricated mask, resulting in nominally identical and well-defined Cu nanoparticles in both beds (Fig. 1c–g). The catalyst bed is accessed via an optical microscope (Nikon Eclipse LV150) equipped with a dark-field 50× objective (LU Plan ELWD 50X/0.55). In this way, the individual Cu nanoparticles can be resolved as individual scattering point sources due to localized surface plasmon resonance (LSPR) excitation in the metallic state[33] (Fig. 1f).

**Multiplexed single particle plasmonic nanoimaging.** Dark-field scattering spectroscopy (DFSS) has during the last decade developed into a valuable tool for materials science and (single particle) catalysis[34], due to its comparably simple and easy to use instrumentation[35]. In essence, it relies on an optical microscope coupled to an imaging spectrometer to collect visible light scattering spectra from nanoparticles of interest, if desired, with single nanoparticle resolution. In a typical experiment, changes in the measured spectra, such as the wavelength shift of the scattering peak, are attributed to chemical or structural changes in the individual particles. DFSS traditionally relies on the use of a grating spectrometer to generate wavelength-resolved single particle scattering spectra[35,36], and is therefore limited to

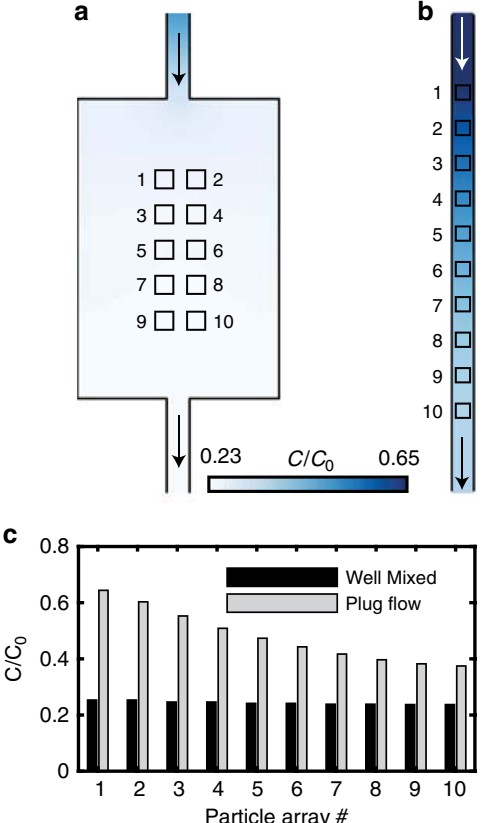

**Fig. 2 Simulated O₂ concentration profiles inside the two catalyst bed geometries.** Concentration profiles presented as colored surfaces for **a** the well-mixed reactor and **b** the plug-flow reactor illustrating the small / large variations in the well-mixed/plug-flow reactor, respectively. **c** The reactant concentration relative to the inlet ($C_0$) at each of the particle array patches for the two designs. Numbers on the x-axis correspond to the numbers in (**a**) and (**b**). In all simulations, the active catalyst is simulated as patches on the wall with a first-order surface reaction, the rate of which (same for both reactor geometries) is chosen so as to match the experimentally observed global conversion over the reactor (see Supplementary Methods for details). The flow through the system was set to $1.5 \times 10^{12}\,\text{s}^{-1}$, based on experimentally measured flow rates.

measuring a few tens of particles simultaneously[35] (with one exception where almost 100 particles could be measured[37]). To increase the number of simultaneously measured particles, so-called hyperspectral spectrometers have been developed[35]. They enable the collection of a combination of 2D images and 1D spectral information in each pixel. However, since this technique relies on taking several 2D images at different wavelengths, temporal resolution scales linearly with spectral resolution and higher intensity light sources are required. By relying on a single wavelength for excitation and readout, both temporal resolution and the number of measured particles can be maximized, which has been utilized to study hundreds of single nanoparticles in parallel[38]. Similarly, in the present study, we have relied on white light imaging to enable multiplexed readout of potentially thousands of single nanoparticles simultaneously. In this solution, which we call multiplexed single particle plasmonic nanoimaging, optical contrast is generated by measuring changes in the scattering intensity in the whole visible range from each of the single nanoparticles.

Specifically, for the Cu catalyst system in the present work, we rely on the large optical contrast between metallic and oxidized Cu as indicator for the oxidation state of the individual

particles[28,39–42], which we measure in situ for all of the 1000 nanoparticles in our two types of catalyst beds (Fig. 3). The optical response from Cu nanoparticles has previously been used to study the oxidation of both particle ensembles[41–43] and individual nanoparticles[39] and to study the state of Cu nanoparticle catalysts[25,28]. To demonstrate this approach for a single Cu nanoparticle with the same dimensions as described above, we have oxidized it by exposure to a flow of 0.1% O₂ in Ar at 200 °C, while continuously monitoring its scattering spectrum (Fig. 3a). The spectral evolution is in very good agreement with a corresponding study of single Cu nanoparticle oxidation by Nilsson et al.[39]. By then integrating the scattering intensity over the whole measured wavelength range, the total scattering intensity is found at each time step, and plotted as function of time in Fig. 3b. Clearly, this integrated scattering intensity exhibits an almost monotonic decrease over time during the exposure to oxygen (gray background) until a new equilibrium state is reached at the end of the oxidation process. The slight minimum at t ~ 150 s is the consequence of the small increase in scattering intensity in the 500–700 nm spectral range observed during the final phase of the oxidation. It can tentatively be explained by a structural change of the particle in the oxide phase or a change in the oxidation state of the oxide as proposed by Bu et al.[28]. We can, therefore, use the integrated scattering intensity signal from a single Cu nanoparticle as a direct indicator of its state of oxidation. Accordingly, the same effect can be observed in a dark-field microscope image of a 100 nanoparticle array patch (Fig. 3c, d), where the Cu particles in the metallic state appear brighter than the same particles after being exposed to an oxidizing environment (400 °C, 0.1% O₂ in Ar). The observed somewhat different absolute scattering intensities exhibited by different individuals in the array are the consequence of slight variations in particle size and shape after recrystallization induced by the high operating temperature[44] (Fig. 1g, h and Supplementary Fig. 5), and are preserved also after oxidation. This is in good agreement with finite-difference time-domain (FDTD—details in "Methods") simulations of the scattering intensity from particles with randomized size and shape variations in a 100 particle array patch, as they are converted from metallic Cu to Cu₂O (Fig. 3e, f).

**Operando experiments focusing on Cu nanoparticle patches.** The first catalytic experiment was carried out by heating the well-mixed reactor to 400 °C at a constant flow of 7% CO in Ar at an inlet pressure of 4 bar. We then gradually increased the incident O₂ concentration from 0 to 0.5%, in steps of 0.02% (Fig. 4a). During the first 4 h of the experiment, the online QMS measurement reveals a linear increase in CO₂ production as O₂ concentration is increased in the feed, but only a minor increase in the outlet O₂ concentration. In this regime, the O₂ conversion is approximately 80%. Close to the 4 h mark, at an O₂ feed concentration of 0.3%, a significant decrease in CO₂ and an increase in O₂ is observed, indicating a significant loss of catalyst activity. After this rapid initial activity loss, a continued increase in incident O₂ results in a further, but significantly slower, decrease in CO₂ production down to a level of O₂ conversion of only ca. 16%. Close to the 7 h mark, we reverted the experiment by stepwise decreasing the inlet O₂ concentration. This results in a low catalyst activity down to an inlet concentration of 0.2% O₂, at which we observe the rapid recovery of activity back to the same level as observed during the increase of O₂.

To correlate the measured activity with the state of the catalyst, we utilize the optical response from each particle patch (blue lines in Fig. 4b). As the main result, we observe a slow and continuous decrease of patch scattering intensity during the first 4 h of the experiment, followed by a more dramatic loss of scattering

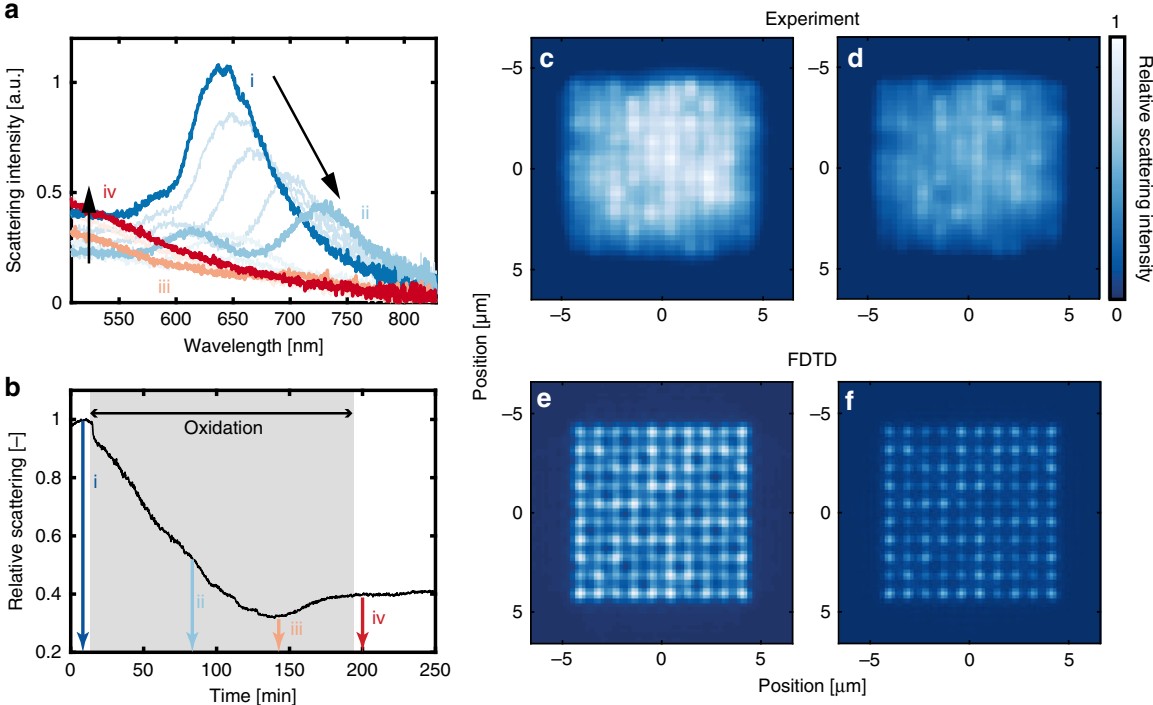

**Fig. 3 Optical response of individual Cu nanoparticles during oxidation. a** Scattering spectra from a Cu nanoparticle during oxidation in 0.1% $O_2$ in Ar carrier gas at 200 °C. Four stages of oxidation are highlighted: (i) metallic, (ii, iii) partly oxidized, (iv) bulk oxidized. The arrows indicate how the spectra evolve when going from stage i to ii (right arrow) and from stage iii to iv (left arrow). **b** Total scattering intensity integrated across the entire wavelength range of 500–850 nm for the same particle as in (**a**) during oxidation in $O_2$ (shaded area). The three colored arrows correspond to the time of the spectral snapshots presented in (**a**). **c, d** Dark-field scattering microscopy image of a 100 Cu nanoparticle array before (**c**) and after (**d**) oxidation. **e, f** FDTD simulated intensity profiles of a 100 Cu nanoparticle array with all particles simulated as metallic Cu (**e**) or $Cu_2O$ (**f**). The radius of the simulated Cu nanoparticles was randomized (±5 nm) to illustrate the effect of a small size difference on the intensity profiles. The images in (**c–f**) have been normalized to the same scattering intensity levels.

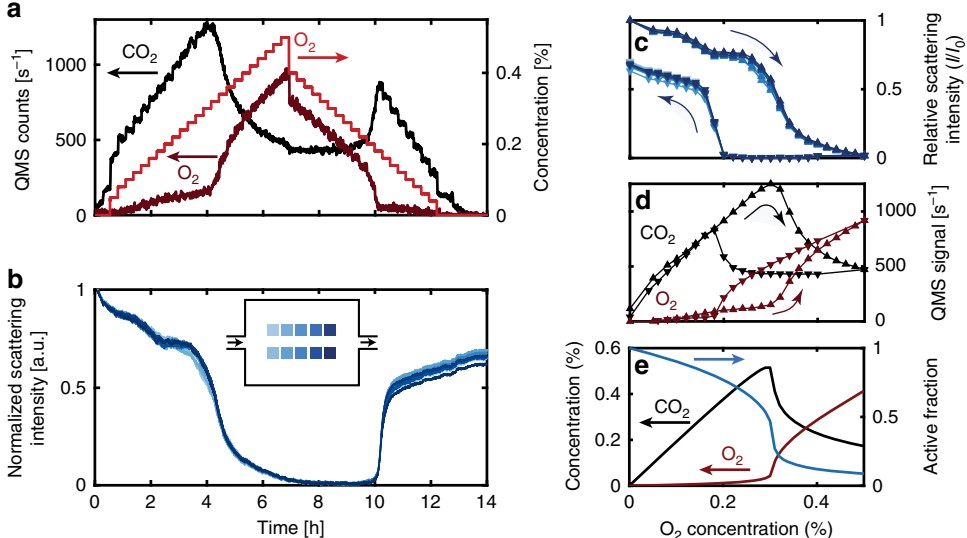

**Fig. 4 CO oxidation experiments over 1000 Cu nanoparticles in the well-mixed reactor.** During the entire experiment, the nominal inlet CO concentration was held constant at 7% and Ar was used as the carrier gas and the pressure at the inlet was 4 bar. **a** Time trace of the nominal $O_2$ inlet concentration in the gas feed (red) and of the outlet concentration of $CO_2$ (black) and $O_2$ (dark red) measured by the QMS. **b** Scattering intensity time evolution of the ten patches, each comprised of 100 nominally identical Cu nanoparticles. Inset is a schematic of the reactor with the nanoparticle array patches color-coded in blue with respect to their position in the bed. Arrows indicate flow direction. **c** Patch scattering intensity plotted as a function of nominal inlet $O_2$ concentration during the increase and decrease of the $O_2$ concentration. **d** Measured $CO_2$ and $O_2$ outlet concentrations plotted as a function of the nominal inlet $O_2$ concentration during the $O_2$ concentration increase (upward triangles) and decrease (downward triangles), respectively. **e** Simulated $CO_2$ production and active fraction of the catalyst calculated with a microkinetic model for a single continuously stirred tank reactor (CSTR). We note that the x-axis in (**c–e**) is the same.

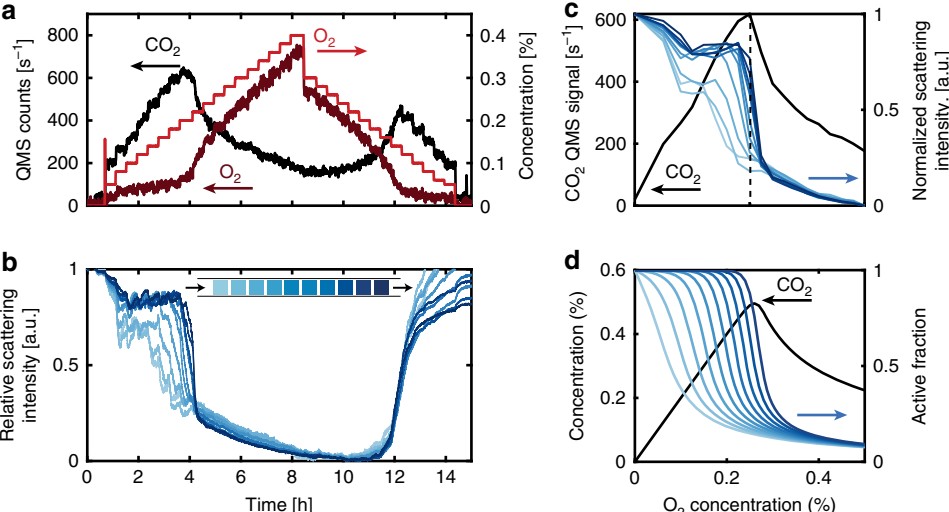

**Fig. 5 CO oxidation experiments over Cu catalysts in the plug-flow reactor.** During the entire experiment, the nominal inlet CO concentration was held constant at 7%, Ar was used as the carrier gas and the pressure at the inlet was 4 bar. **a** Time trace of the nominal $O_2$ inlet concentration in the gas feed (red) and of the outlet concentration of $CO_2$ (black) and $O_2$ (dark red) measured by the QMS. **b** Integrated scattering intensity time evolution of the ten Cu nanoparticle patches, each comprised of 100 nanoparticles. Inset is a schematic of the reactor with the nanoparticle array patches color-coded in blue with respect to their position in the bed. Arrows indicate flow direction. **c** Measured $CO_2$ outlet concentration (black) and integrated patch scattering intensity (blue) plotted as a function of the nominal inlet $O_2$ concentration during $O_2$ concentration increase in the feed. **d** $CO_2$ production and active catalyst fraction simulated with a microkinetic model using ten CSTRs in series.

intensity, which coincides with the loss of activity measured by the QMS (Fig. 4c, d). This is similar to the observations made in a study performed on a much larger model catalyst[28], indicating that our results are not only related to the specific reactor geometry. Furthermore, the optical response from the ten patches is essentially identical and thus corroborates the finite-volume simulations that predict very small reactant gradient formation only (cf. Fig. 2a, c).

To understand the evolution of the scattering intensity signal, we remind ourselves that a loss in scattering intensity is characteristic for the oxidation of the Cu particles because metallic Cu loses its intrinsic LSPR, and thus its ability to efficiently scatter visible light, when it becomes oxidized (cf. Fig. 3)[28,41,42]. Therefore, as a first conclusion from these measurements, we find that bulk oxidation of Cu nanoparticles is related to a drastic activity drop in the CO oxidation reaction, in agreement with the previous work[28]. To this end, XPS measurements performed on a sample analog exposed to the same reaction conditions show that the final oxidized state of the Cu nanoparticles is predominantly $Cu_2O$ (Supplementary Fig. 6).

The initial period of a slowly decreasing scattering intensity can be explained by the gradual formation of a (surface) oxide, starting as soon as $O_2$ is introduced, and during which the apparent catalyst activity increases due to the increase in available oxygen (Fig. 4a–d). In contrast, introducing $O_2$ alone (without CO) results in an equivalent scattering intensity drop that occurs within ca. 3 min only (Supplementary Fig. 7). Hence, our results directly visualize the nature of a process where two competing reactions occur simultaneously, i.e., Cu oxide formation and conversion of CO to $CO_2$, whose rates both rely on the availability of oxygen. This is in line with the further observation that the gradual decrease in activity and scattering efficiency of the nanoparticles spans across several $O_2$ concentration steps (Fig. 4c, d), and further confirms that, as the Cu particles become more oxidized, their activity decreases. However, we also note that even a fully bulk oxidized catalyst bed retains approximately 20% of its activity (cf. Fig. 4a in the interval 4–8 h; for empty reactor control, see Supplementary Fig. 8).

Executing the same analysis when decreasing the $O_2$ concentration, we find that the recovery of the scattering intensity takes place at a lower nominal feed $O_2$ concentration compared to the preceding oxidation. This reveals hysteresis between the oxidation/reduction thresholds and thus for catalyst activity with respect to $O_2$ concentration in the feed (Fig. 4c, d).

To further analyze these results, we constructed a microkinetic model based on continuously stirred tank reactors (CSTRs—details in "Methods"). The model describes both CO oxidation, informed by first-principles calculations, and loss of catalyst activity due to oxidation of the active sites. We modeled the well-mixed reactor as a single CSTR, informed by the results from the finite-volume simulations. Furthermore, we selected the kinetic parameters for oxidation of the active sites and reaction with the oxide such that the simulation predicts a loss of activity at the same inlet $O_2$ concentration and a similarly reduced conversion under significant oxidation as observed experimentally. In this way, we observe a behavior qualitatively consistent with the experiment (Fig. 4e). Interestingly, the simulation then also suggests that, at around the critical inlet $O_2$ concentration, the catalyst oxidation accelerates as the $CO_2$ production rate drops.

As the next step, we turn to the plug-flow reactor and perform the same type of experiment. As for the well-mixed reactor, stepwise increasing the $O_2$ concentration at constant nominal 7% CO inlet concentration results in linearly increasing $CO_2$ production until approximately 0.2% $O_2$, where the $CO_2$ production drops dramatically and an increased amount of $O_2$ is measured exiting the reactor (Fig. 5a). Turning to the optical response, we immediately notice significant differences compared to the well-mixed reactor, namely that the characteristic scattering intensity decrease due to Cu particle oxidation does not occur simultaneously on all array patches (Fig. 5b, c). Instead, the patches closest to the inlet start to oxidize several hours before, as well as oxidize more slowly, than the ones toward the outlet of the reactor. Furthermore, there is a gradual progression of the bulk oxidation onset point with respect to nominal inlet $O_2$ concentration throughout the reactor, where the different catalyst patches oxidize in a range from 0.12 (close to the inlet) to 0.2%

(close to the outlet) $O_2$ (Fig. 5c). Interestingly, the main activity drop measured by the QMS occurs first when the last few patches undergo complete oxidation according to the scattering signal. In other words, from the integrated QMS response one observes a continued apparent increase in catalyst activity despite >50% of the bed already having been converted to the less active bulk oxide phase.

The observed specific delay in bulk oxidation of each patch, as manifested in the plug-flow reactor but absent in the well-mixed one, can be explained by considering the local $O_2$ concentration generated by reactant conversion and predicted by the simulated concentration profiles (cf. Fig. 2). Focusing first on the well-mixed reactor, where the local concentration at the particles is considered the same throughout the reactor (cf. Fig. 2a), we note that the main oxidation event (steepest decline in intensity) is observed at an inlet $O_2$ concentration of 0.3%. Locally, inside the well-mixed reactor, this corresponds to an $O_2$ concentration of 0.072% (details of how this number was calculated can be found in the Supplementary Methods), which corresponds to the true critical $O_2$ concentration that induces bulk oxidation of the Cu particles in the system at hand. Turning to the plug-flow reactor, we observe bulk oxidation of the first patch at an inlet $O_2$ concentration of 0.12%. Translated to the local concentration based on the simulated concentration profiles presented in Fig. 2b, the first patch oxidizes at a local $O_2$ concentration of 0.074%. In other words, we observe essentially the same local $O_2$ concentration required for bulk oxidation in both reactor types. This is an important result because it (i) shows that the Cu particles in both reactors have identical properties and (ii) validates the simulation model.

Finally, after the main scattering intensity drop has occurred for all patches in the plug-flow reactor, we observe a gradual decrease in the $CO_2$ production and a corresponding small gradual, and for all patches now again essentially identical, decrease of the optical scattering intensity as the $O_2$ concentration in the flow is further increased (Fig. 5a, b). In analogy to the well-mixed reactor, here we also note that even a fully bulk oxidized catalyst bed retains an $O_2$ conversion of ca. 10%. When decreasing the $O_2$ concentration, we observe the simultaneous recovery of scattering intensity of all patches at an inlet $O_2$ concentration of 0.12%, which coincides well with the recovery of the catalytic activity measured by the QMS (Fig. 5b). This is in good agreement with what we observed in the well-mixed reactor and thus with the overall scenario since the rate of Cu oxide reduction is not limited by the availability of CO as it is present in excess.

To further corroborate these findings, we used the same microkinetic model as above, with the Cu oxidation and oxide-consuming kinetic parameters determined for the well-mixed reactor, but using ten CSTRs in series, each tank corresponding to one nanoparticle patch. The simulation predicts that the patches will oxidize sequentially, whereas the drop in $CO_2$ production is only observed at the reactor outlet once all patches have lost significant activity (Fig. 5d, details in Supplementary Methods), in very good agreement with the experiment.

For an intermediate summary, we note first that our measurements, which spatially resolve the whole catalyst bed, can be used to visualize the impact of reactor geometry on local reactant conversion and transport and how very different and contradicting conclusions about catalyst active phase can be drawn for the same catalyst material depending on reactor design. In the plug-flow geometry, although a majority of the catalyst patches appeared oxidized, the measured overall activity remained high or even kept increasing. For the well-mixed reactor on the other hand, bulk oxidation of all patches occurred simultaneously and coincided with a massive loss of activity,

indicating that the bulk oxide state is not very active. Furthermore, in the plug-flow geometry, probing the state of the catalyst only near the entrance of the reactor would lead to the immediate conclusion that also fully oxidized particles retain their high activity. In contrast, measuring the catalyst state only near the outlet would indicate that a loss of activity is closely correlated with bulk oxidation.

As the second main aspect, in both reactors, we see indications of surface oxide formation, manifested as continuous decrease in scattering intensity, already at low oxygen concentrations where no decrease in catalytic activity is measured by the QMS. This can be interpreted as an indication that a mixed metal-(surface)oxide state is the most active phase of Cu for the oxidation of CO at the present reaction conditions, which has been suggested previously in an ambient-pressure XPS study[29] and by environmental TEM[31]. At the same time, our measurements also show that probing the surface state of the catalyst alone is insufficient when determining the active phase of a particle as we see a clear change in activity not only initially when only the surface is oxidized but an even more dramatic change when bulk oxidation occurs.

**Operando experiments focusing on individual Cu nanoparticles.** Having established these intermediate conclusions at the patch level, we now analyze the same experimental data with single particle resolution by tracking scattering intensity changes for the entire set of 1000 nanoparticles for the well-mixed reactor (Fig. 6a, b) and the plug-flow reactor (Fig. 6c, d and Supplementary Movie 1), by plotting them as function of time together with the QMS response. For the well-mixed reactor, we immediately recognise the same general trend identified in Fig. 4 for the patches, namely that bulk oxidation—on average—takes place simultaneously for all patches. Similarly, also for the plug-flow reactor, we see the same general trend identified in Fig. 5 for the patches. However, and as the key point, we also notice a very significant spread in the single particle response on the level of several hours for some of the patches.

To capture and quantify the differences between individual particles, we define and extract a characteristic oxidation time, $\tau_{ox}$, as the time when the normalized scattering intensity of a particle has reduced to 50% of its maximum value (orange marks in Fig. 6b, d), as illustrated for a selected number of individual particle response traces (Fig. 7) and statistically analyzed for the complete set of 1000 particles (Fig. 8) for both reactor types. First focusing on the single particle traces (Fig. 7) it becomes clear that they can be very different. For example, some particles in both reactors follow the average optical response closely and exhibit monotonous optical response (Fig. 7d), while others deviate significantly. For example, we find recovered intensity after an initial decrease (Fig. 7a, c, e, g), which we interpret as transient reduction of the particle to a metallic state before bulk oxidation, as well as a distinct and repeated blinking in scattering intensity (Fig. 7e, g), which we interpret as a dynamic switching between a reduced and oxidized state. Similar behavior has previously been observed under oxygen-lean conditions (CO:$O_2$ > 3:1) studied by AP-STM[15], where metallic Cu islands were formed through local reduction of $Cu_2O$. This single-particle specific behavior can be understood as the consequence of each particle having a unique surface structure (facets), morphology (grains) and other defect abundance, which mediates both the $O_2$ and CO adsorption affinity to the particle[17,45], as well as the critical local $O_2$ concentration needed to initiate the bulk oxidation process, in analogy to observations for the hydride formation in individual Pd nanoparticles[44].

Now analyzing the single-particle $\tau_{ox}$-statistics, for the well-mixed reactor, we see a distribution, which peaks after about 4.2 h

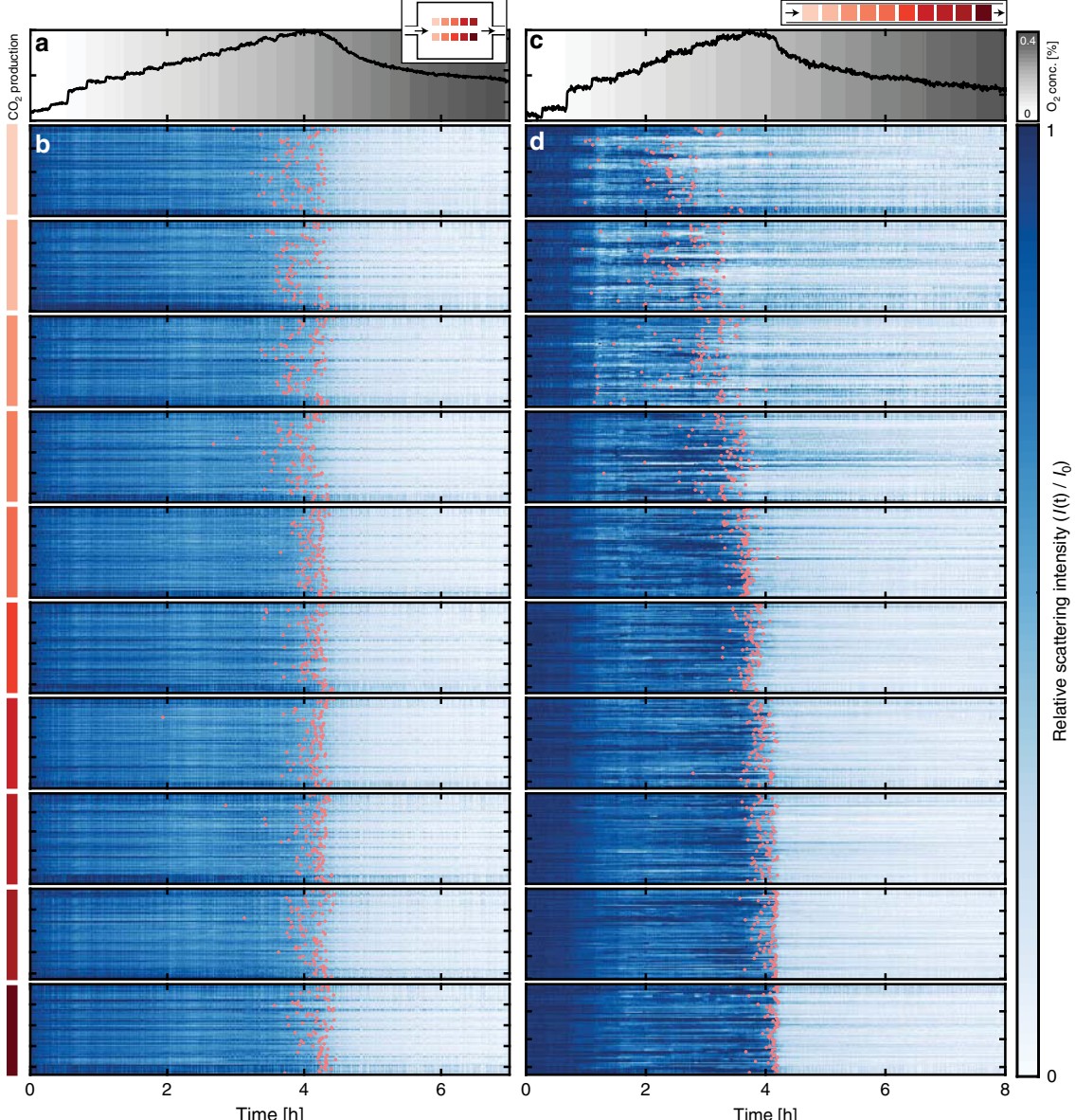

**Fig. 6 Individual optical response from all 1000 Cu particles. a** Integrated $CO_2$ production measured by the QMS from the well-mixed reactor during the stepwise increase of nominal inlet $O_2$ concentration depicted as the shaded background. The inset depicts the reactor geometry and indicates the position of the color-coded particle patches. Arrows depict reactant flow direction. **b** Scattering intensity time evolution from the individual nanoparticles in each patch (color-coded to the left). Each patch subset contains ~100 individual lines corresponding to a single particle in the patch. The orange markers indicate the oxidation time, $\tau_{ox}$, defined as the time when each particle's scattering intensity has decreased to 50% of the initial level. **c, d** The same as (**a, b**) but for the plug-flow reactor. The scattering intensity in each individual particle trace has been normalized to its initial intensity and minimum intensity after oxidation, to facilitate direct comparison between particles.

for all patches (Fig. 8a, b). We also see that the distribution is quite broad, and broadest for patches closest to the inlet. In the most extreme case, within one patch, individual particles exhibit $\tau_{ox}$ that range from 2.8 to 4.4 h (based on the whiskers in the box plots to not overemphasize the few even more extremely spread particles). We also observe a slight position dependence on the $\tau_{ox}$ where particles placed further up-stream oxidize prior to particles further downstream. This indicates that we have a slight PFR-like behavior also in the well-mixed reactor, as also predicted by our model (Fig. 2a, c). Furthermore, we find that a sizable fraction of single particles is considered oxidized (i) before the QMS signal indicates a loss in catalytic activity (ca. 90%, Supplementary Fig. 9a) and (ii) before the average optical signal indicates bulk oxidation (ca. 50%, Supplementary Fig. 9b). For the

reversed process of particle reduction when decreasing the $O_2$ concentration in the feed, we find very narrow distributions in all patches which peak simultaneously, irrespective of position in the reactor (Supplementary Fig. 10a, b) that can be explained by the excess of the reducing agent CO.

Turning to the plug-flow reactor, we find drastic single-particle effects, which in magnitude strongly depend on the patch and thus position along the reactor (Fig. 8c, d). For the most extreme case of the most upstream patch, $\tau_{ox}$ of the individual particles is spread out over almost 3 h, whereas within the array closest to the outlet all particles bulk oxidize within only 25 min (based on whiskers in Fig. 8d). The narrower $\tau_{ox}$ distribution observed for particles oxidized later can be rationalized by remembering that the $O_2$ concentration is increased during the experiment and that

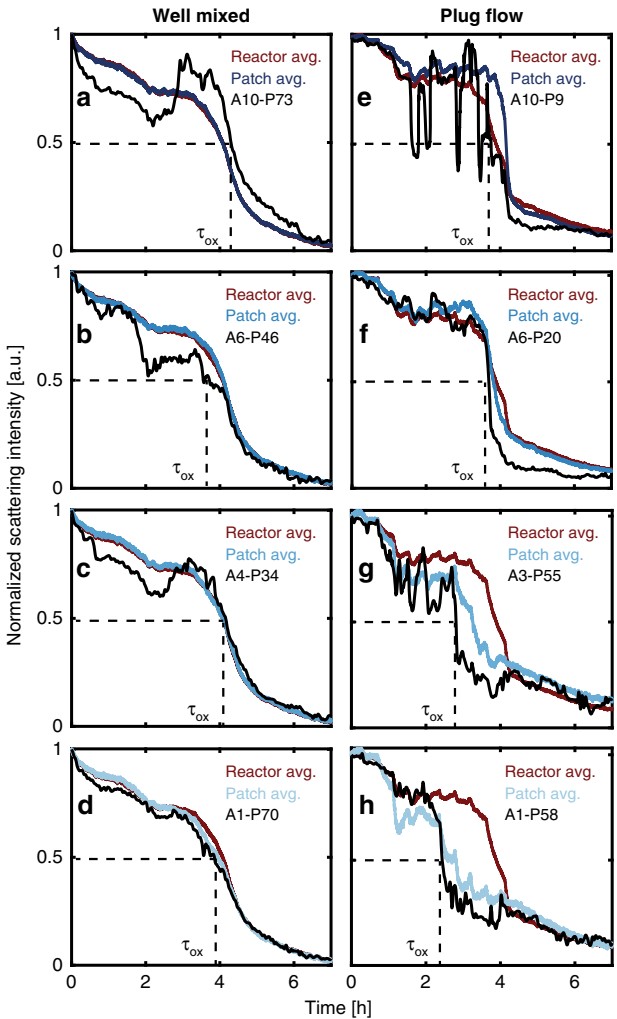

**Fig. 7 Optical response of selected individual Cu particles.** Four individual particle responses from the well-mixed (**a–d**) and the plug-flow (**e–h**) reactor. Red lines correspond to each reactor average scattering intensity, blue lines to the relevant patch average and the black lines to single particle responses. The single particle legends correspond to the patch (A) (cf. Fig. 2a, b) and particle number (P) within the patch.

fewer particles are actively consuming $O_2$ due to deactivation. Both these effects contribute to a larger $O_2$ excess available for particles placed further downstream and consequently a faster (less mass transport limited) Cu oxidation (see Supplementary Discussion for details).

Comparing the $\tau_{ox}$ distribution of the individual particles in each patch (Fig. 8d) with the averaged scattering response from each patch (Fig. 8c) reveals that the steplike and fluctuating average response is a consequence of the individual particles in the patch oxidizing at very different and particle-specific times rather than a collective behavior of all particles within the patch (see Supplementary Discussion for details). Consequently, utilizing the information extracted from a single particle can lead to significantly different conclusions than the patch-averaged response with respect to the catalyst state and active phase. Similar to the well-mixed reactor, the decrease in $CO_2$ production is detected after a majority of the individual particles are considered oxidized (Supplementary Fig. 9c, d) and the subsequent reduction during the decrease in $O_2$ concentration occurs in a narrow time interval (Supplementary Fig. 10c, d).

**Catalyst oxidation dynamics and active phase**. In the literature, different scenarios for the CO oxidation reaction mechanism and the active phase have been described, ranging from a Langmuir–Hinshelwood mechanism assuming surface oxygen without Cu oxide formation[30], to a Mars-Van Krevelen (MvK) mechanism where oxygen in the oxide is consumed via a reaction with CO resulting in transient reduction to a lower oxidation state or to metallic Cu[46], to a crystalline $CuO_x$ phase reversibly transforming into an amorphous highly active oxide phase[31]. In this respect, our experiments show that at the present operando conditions of ca. 2 bar in the reaction zone, formation of a surface oxide starts immediately upon admission of $O_2$ to the feed (manifested as small decrease in scattering intensity in the initial phase of the experiments) and occurs in the regime where the catalyst is highly active. Significant deactivation becomes apparent only once bulk oxidation to $Cu_2O$ has occurred, irrespective of reactor design. This highlights that tracking the surface oxidation state alone, as for instance with XPS, may be insufficient to understand the active phase for Cu catalysts and other metals with the potential for bulk oxidation. Furthermore, this agrees well with previous research identifying a pure bulk oxide state as less active than a bulk metallic state[15,28]. At the same time, we also note that it ultimately is the surface of the particle on which the reaction occurs and thus deactivation most likely occurs already prior to completed oxidation of the bulk of the particle.

Secondly, our experiments also resolve the highly dynamic nature, as well as high single-particle specificity, of the catalyst in its most active state, that is, prior to bulk oxidation. Specifically, this is manifested in that a large fraction (up to ca. 90%, Supplementary Fig. 11) of the individual nanoparticles exhibit partial reduction of the surface oxide during reaction or even dynamic oscillations between an oxidized and reduced state in the range of highest catalyst activity (cf. Fig. 7). This is a clear indication for a process where the surface oxide is transiently and dynamically consumed by CO to form $CO_2$. In combination, these findings thus suggest that a surface-oxidized catalyst is present in the state of highest catalyst activity, and that a dynamic mix of metallic Cu and surface oxide constitutes the most active phase on most nanoparticles. However, we also highlight that this scenario is not observed in all individual particles, since a small fraction (cf. Fig. 7d) does not exhibit the clear signature of dynamic oxidation/reaction but exhibits a more monotonous oxidation, presumably controlled by the particle-specific morphology and defect structure. This highlights a possible shortcoming of single point probes with single particle resolution, if not a large number of particles are studied to generate a statistically relevant number of single particle responses.

Along the same lines, we highlight the observation that multiple different catalyst states exist dynamically across the catalyst bed, both depending on the position in the bed and the individual structural properties of the single nanoparticle. Hence, we propose that this dynamic coexistence of multiple states within the catalyst bed and between individual particles, as well as the observed dynamic fluctuations between states that are readily averaged out in any ensemble measurement, is one of the main contributing factors to contradicting reports about the most active oxidation state of Cu in the CO oxidation reaction[15,26–29].

## Discussion

The development of experimental methodologies that enable operando studies of heterogeneous catalysts with single nanoparticle resolution, while at the same time being able to probe an entire catalyst bed comprised of a relevant ensemble of particles, is identified as an important enabling technology for further developing our understanding of catalyst systems. In this work,

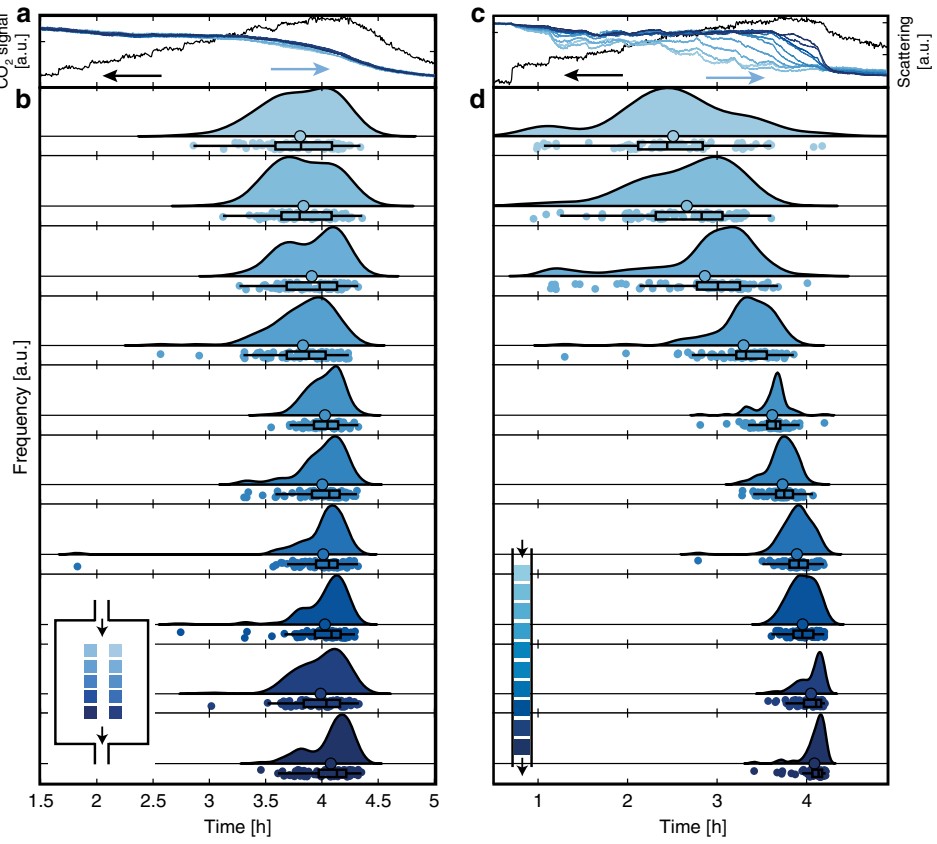

**Fig. 8 Single Cu particle oxidation time analysis for the two reactors. a** $CO_2$ production measured by the QMS at the outlet of the well-mixed reactor (black, left axis) together with the integrated scattering intensity for each of the ten patches (blue, right axis). **b** Oxidation times for the individual particles in the well-mixed reactor, presented as half violin plots. Each distribution corresponds to data collected from the 100 single particles contained within a patch, with each patch outlined and color-coded in the inset. Below each of the distributions are the individual particle data points presented as colored dots, the mean oxidation time (outlined large circle) and a box plot showing the median oxidation time, the inner percentile (box) and whiskers corresponding to the lower/upper adjacent values (horizontal line). **c**, **d** Same as (**a**, **b**) but for the plug-flow reactor. Note the slightly different scale on the *x*-axis.

we demonstrated how a combination of nanofluidic reactor technology with highly multiplexed single particle plasmonic nanoimaging can be used for such studies. Specifically, our platform enabled catalyst state characterization with single particle resolution from a set of 1000 nanoparticles in combination with online QMS analysis of activity from the same particle set.

Using this platform, we addressed the CO oxidation reaction over 1000 Cu nanoparticles placed in two nanoreactors with different designs (one with significant reactant concentration gradients and one without), in an attempt to understand the origin of the contradicting information about the active phase found in the literature. As the first stage, we demonstrated how reactor geometry controls the local reactant distribution and how this results in large spatial variations of the reaction conditions in the catalyst bed, which, in turn, leads to dramatic spatial variations of the oxidation state across the catalyst bed. This effect, we identified as a key reason for potentially erroneous structure-function relationships if only an averaging (e.g., XPS), or only a high-resolution single point (e.g., TEM) experimental probe, is utilized.

Secondly, by studying the entire catalyst bed with single nanoparticle resolution, we revealed that the oxidation state dynamics of single Cu particles can differ drastically between individuals, and that these single particle dynamics, which we attribute to structural/morphological differences at the single particle level, are not even resolved when averaging an ensemble as small as 100 particles. Furthermore, we observed a highly

dynamic oxidation state of the single particles in the regime where the catalyst is found most active, in its most extreme manifestation oscillating between reduced and (partly) oxidized. This suggests that a dynamic mix of metallic Cu and surface oxide constitutes the most active catalyst phase.

In conclusion, these results clearly show that neither a single point measurement nor a measurement that averages the whole or even only a smaller fraction of the entire catalyst bed will give the true picture of the catalyst in terms of state and activity dynamics. In this way, our results highlight the importance of experimental methods that enable the multiplexed simultaneous single particle probing of catalyst state across the entire catalyst bed that hosts a number of particles large enough to constitute a relevant ensemble. To this end, the method presented here is currently limited to particles with a size >50 nm in at least one dimension in order to assure a detectable optical contrast from the singe catalyst nanoparticles. Since this is one order of magnitude larger than technically used catalyst nanoparticles, which typically are in the sub-10 nm size range, it is of interest to implement strategies like indirect nanoplasmonic sensing to enable the study of catalyst nanoparticles in this size regime in the future[47–49]. Furthermore, we envision our concept being implemented in parallel with methods that feature atomic resolution (e.g., environmental TEM) to (i) identify interesting individuals from a large ensemble and (ii) verify that a particle monitored by a potentially invasive technique is not behaving significantly different compared to the rest of the ensemble.

## Methods

**Experimental setup**. Experiments were carried out in our recently developed nanoreactor platform[25] (Supplementary Fig. 3) consisting of a micro-/nano- fabricated Si/SiO$_2$ chip that acts as a chemical reactor for gas phase reactions. The chip is separated into three main parts, the inlet microchannel system, designed as a U-shape (Fig. 1a), the nanoreactor containing the active catalyst (Fig. 1b) and the outlet microchannel that connects the outlet of the nanoreactor to a QMS (Hiden HAL/3F PIC). The U-shape design of the inlet microchannel enables rapid exchange of the admitted gas (<30 s) via an array of mass flow controllers (MFCs— Bronckhorst Low ΔP) that are used in combination with a pressure controller (Bronckhorst) to set the gas composition and enable QMS operation at up to 10 bar. On the back of the chip, a Pt thin film heater and resistive thermometer is placed and the on-chip temperature is controlled using a PID (Lakeshore 335) to enable precise temperature control of the reaction zone of the chip in the range from 20 °C to 450 °C. The chip is mounted under an upright optical microscope (Nikon LV150) in a specifically designed chip holder that acts as an interface for the gas supply and electrical connections (Supplementary Fig. 3). The microscope is connected to an imaging system that allows for imaging and spectroscopy of the catalyst particles placed inside the reactor using an long working distance dark-field objective (Nikon LU Plan ELWD 50X/0.55), a grating based spectrometer (Andor Kymera i193) and a EMCCD camera (Andor iXon 888). The light source is a 50 W halogen lamp (NikonLV-HL50W LL).

For the CO oxidation experiments, ultrapure CO (10% in Ar) and O$_2$ (2% in Ar) were used with Ar carrier gas (99.99999% purity) and fed with different concentrations into the chip. The inlet pressure was set to 4 bar and a total flow of 10 ml/min through the microchannels was used.

**Nanofabrication**. Nanofabrication was carried out in cleanroom facilities of Fed. Std.209 E Class 10–100, using electron-beam lithography (JBX-9300FS / JEOL Ltd), direct-laser lithography (Heidelberg Instruments DWL 2000), photolithography (MA 6/Suss MicroTec), reactive-ion etching (Plasmalab 100 ICP180/Oxford Plasma Technology and STS ICP), electron-beam evaporation (PVD 225/Lesker), magnetron sputtering (MS150/FHR), deep reactive-ion etching (STS ICP/STS) and wet oxidation (wet oxidation/Centrotherm), fusion bonding (AWF 12/65/Lenton), and dicing (DAD3350/Disco). To achieve precise alignment for consecutive lithography steps, alignment marks, fabricated using electron beam lithography (EBL) or photolithography (PL), were used. These marks allowed us to achieve a positioning error of approximately 10 nm using EBL and 1 mm using PL. A detailed description of the fabrication steps can be found in the Supplementary Methods and Supplementary Fig. 2.

**Transmission electron microscopy**. The TEM images were acquired by a FEI Tecnai T20 (LaB6 filament) operated at 200 kV. The images were taken in bright field-mode at 71 or 43 kX magnification. An objective aperture was inserted to remove ghost images due to the diffraction in the crystal planes of the particles.

**X-ray photoelectron spectroscopy**. XPS measurements were performed with a PHI 5000 VersaProbe III (Physical Electronics). The excitation source was monochromatized $K_\alpha$-line of aluminum operated at 50 W. All spectra were recorded with an energy step width of 0.1 eV and a pass energy of 55 eV. The base pressure was always lower than $5.0 \times 10^{-7}$ Pa. All spectra were corrected by setting the adventitious C-1s peak of the C–C bond to 284.8 eV.

**Optical data collection and processing**. Videos of the catalyst bed were collected by an upright optical dark-field microscope (Nikon Eclipse LV150) with a 50× objective (Nikon LU Plan ELWD 50X/0.55) equipped with an imaging EMCCD camera (Andor iXon 888). This setup resulted in each pixel corresponding to an area of 250 × 250 nm on the sample. Images were collected with an integration time of 5 s. Due to slight mechanical instabilities, mainly due to thermal fluctuations of the chip, post processing stabilization of the images was performed using image registration in Matlabs imaging toolbox (details and code can be found in Supplementary Methods). The individual particle positions were found by applying a wavelet filter (details and code in Supplementary Methods) to the images and utilizing the filtered image with the highest particle contrast and finding the corresponding positions. Individual particle scattering intensities were then calculated as the sum of scattering from 3 × 3 pixels around each individual particles' position (corresponding to an area of 750 × 750 nm on the sample). In this way, the light scattered from each particle could be collected without registering light scattered from neighboring particles. The scattering intensity for particle i ($I_i$) at every time step was then calculated as $I_i = S_i - I_b$, where $S_i$ is the raw scattering intensity and $I_b$ is the dark current collected by the CCD without any incident light. To be able to compare traces from particles with different scattering intensity they were individually normalized as:

$$\widetilde{I}_i = \frac{I_i - I_{i,0}}{I_{i,0} - \min(I_i)} \tag{1}$$

Where $\widetilde{I}_i$ is the normalized scattering intensity of particle i, $I_{i,0}$ is the raw intensity at time 0 and $\min(I_i)$ is the minimum value of $I_i$ during the experiment. To collect the scattering intensity from patches, a sum of the scattering intensity from an area covering the whole patch was collected. For inter-patch comparisons, the same normalization as for the single particle traces was utilized.

**Finite-volume simulations**. Finite-volume simulations were performed using ANSYS Fluent 2019 R3 to predict the reactant distribution inside the two reactor geometries during a catalytic reaction between CO and O$_2$. In the simulations, CO was considered to be in excess and the limiting reactant was O$_2$. For simplicity the reaction was treated as first order with respect to O$_2$ and reactions only take place on each of the particle arrays shown in Fig. 1c, d (more details in Supplementary Methods). The resulting O$_2$ concentration profiles inside the two reactor geometries are presented in Fig. 2.

**Microkinetic model**. A microkinetic model for CO oxidation on Cu was constructed in a tanks-in-series framework. The model describes CO and O$_2$ adsorption and desorption, O$_2$ dissociation, reaction between adsorbed CO* and O* according to a Langmuir–Hinshelwood mechanism, oxide formation with first-order kinetics and reaction between the oxide and CO*[50] (steps provided in the Supplementary Methods). It was assumed that the reactant adsorption steps were unactivated and that CO$_2$ desorption was fast and irreversible. The kinetic parameters for the forward and reverse reaction steps were taken from Falsig et al.[51], calculated using Density Functional Theory with generalized gradient approximation quality. Kinetic parameters for the rate of oxidation of the active sites and the rate of reaction with the oxide were specified in order to model activity loss at the same inlet O$_2$ concentration leading to a similarly reduced conversion as observed experimentally in the well-mixed system. The tanks-in-series framework uses continuously stirred tank reactors (CSTRs) to represent local flow conditions (more details in the Supplementary Methods). As suggested by the finite-volume simulations, one CSTR is sufficient to model the well-mixed system whilst the plug-flow system is better represented by a sequence of CSTRs to permit concentration gradients. For the plug-flow system, each catalyst patch was modeled as a single CSTR—yielding a 10 CSTR network. The system dimensions and conditions correspond to those defined in the main text, using the pressures from the unified flow model calculations (Supplementary Methods). The density of active sites for each nanoparticle was computed using the nanoparticle geometry and centre-to-centre distance of adjacent Cu atoms. The system of ODEs describing each reactor/ series of reactors was solved using the MATLAB R2018a stiff differential equation solver ode15s using backward differentiation formulas (BDF).

**FDTD Simulations**. Finite-difference time-domain (FDTD) simulations were performed using the the commercial software package FDTD Solutions (Lumerical). The Cu nanoparticle patch was simulated as a square 10 × 10 particle array placed with a pitch of 900 nm placed on a SiO$_2$ substrate. Each particle was simulated as a cylinder with a radius of 45 ± 5 nm and a height of 30 ± 5 nm. The random size factor (±5 nm) was introduced to mimic the size distribution in the real samples. A mesh overlay with a resolution of 2 nm (x, y, z) was defined around each nanoparticle. Oxidized particles were simulated by changing the dielectric function of the disk and increasing the particle size in accordance with the expansion factor for Cu$_2$O (1.68)[41]. The dielectric function for SiO$_2$ was taken from Palik[52], the one for Cu from Hagemann et al.[53] and the one for Cu$_2$O from Tahir et al.[54]. When simulating the optical microscopy images, incident light was simulated by two total-field scattered-field (TFSF) sources with an incident angle of 10° to the normal of the sample plane. The two sources were incident 90° relative to each other to mimic the annular nature of the light source used in the microscope. A near field monitor was placed 2150 nm above the sample and the near field monitor data was decomposed to a series of plane waves using far field projections built in to the FDTD solutions software package. Any of the plane waves with angles outside of the numerical aperture (NA) of the objective were removed and the image was constructed from the remaining light. To mimic white light illumination, the scattering images were calculated as the sum of the intensities from 7 wavelengths in the visible range, i.e., 450, 500, 550, 600, 650, 700, and 750 nm.

## Data availability

The data sets generated during the current study are available from the corresponding author upon reasonable request.

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

## Acknowledgements

This research has received funding from the European Research Council (ERC) under the European Union's Horizon 2020 research and innovation programme (678941/SINCAT) and from the Knut and Alice Wallenberg Foundation project 2015.0055. The finite-volume simulations were enabled by resources provided by the Swedish National Infrastructure for Computing (SNIC) at C3SE partially funded by the Swedish Research Council through grant agreement no. 2016-07213. Part of this work was carried out at the MC2 cleanroom facility and at the Chalmers Materials Analysis Laboratory. We also acknowledge fruitful discussion with Prof. H. Grönbeck.

## Author contributions

D.A. designed and fabricated the nanofluidic chips, performed the experiments, performed FDTD simulations, analyzed the experimental data, made all figures and wrote the first draft of the paper. A.B. performed the pressure profile calculations and the microkinetic modeling. A.H. and H.S. supervised the development of the microkinetic model. H.S. performed the fluid dynamic calculations and simulations. S.N. performed the TEM measurements. C.T. performed the XPS measurements. C.L. co-wrote and edited the paper and supervised the project as a whole. All authors gave feedback on the final paper.

## Funding

## Competing interests

The authors declare no competing interests.
