## [Peer Review File · Nature Communications]

REVIEWER COMMENTS

Reviewer #1 (Remarks to the Author):

The authors David Albinsson et al. used the dark-field microscopy and QMS to study the state of copper under operando conditions. This research bridges the gap between single nanoparticle probing and catalyst-bed-averaging. They provide a great idea to combine the advantages of both dark-field microscopy and QMS to reveal the reaction mechanism during the catalytic reaction in-situ and in real time. The method could make us simultaneously know the relationship between scattering spectra, morphology, products and catalytic activity. A lot of important information has been uncovered, and make us understand the oxidation reaction of CO and copper catalyst. The method in this research is general for many gas phase catalytic reactions and the results are also very important in relative research field. However, there are some problems need to be considered carefully.

1. The abbreviate QMS in abstract need a full name.
2. How to make a distance 900 nm? A sentence is needed to introduce it.
3. There are some problems in Fig.1. First, the font size is too small to read. Second, the cross section needs a sign of width and height. Third, the b needs to illustrate the structure, which keeps the low pressure in QMS. Fourth, the size of e, f is too small. Actually, these two figures are very important. Fifth, this figure needs to be well organized.
4. Since the Cu nanoparticles are immobilized in the cells of the bed, what's the scattering intensity of these cells as a background.
5. In Figure 2, is it possible to make the grids connect together, and then the gradient of oxygen concentration will be more smooth. This only needs a discussion but not an experiment.
6. Fig. 4c-e needs a time axis on top of c. In addition, Fig. 4d is not clearly introduced.
7. The color of flow cell in Fig.6a and c may need to be different from that in Fig.6b and d.
8. Copper (I) and Copper (II) in the catalyst may play different functions in the catalyst. The authors show some oscillation of the scattering intensity. How do the Copper (I) and Copper (II) contribute to the scattering signal. And how do the Copper (I) and Copper (II) affect the catalytic activity?

In summary, this paper was written and organized well. This work could have high impact in single particle research field. I am pleased to support the publication in high impact journal like Nature Communications.

Reviewer #2 (Remarks to the Author):

This is a novel and carefully executed study that will be of interest to researchers in catalysis and also a wide field of researchers beyond catalysis.

The experimental details are provided in great detail as well as the statistical analyses employed.

I only question one conclusion that should be considered in the final version. Heterogeneous catalysis is a surface phenomenon and the manuscript comments about the catalytic relationship with the bulk CuO phase. The bulk phase never directly participates in the catalytic reaction since it is the "surface" of the bulk phase that is involved in the catalytic process. For example, the inactivity of the bulk Cu₂O phase may simply be related to the stabilization of a heavily oxidized surface that does not have any available reduced surface metallic sites for CO adsorption. I think it is important to emphasize the surface oxidation state rather than the bulk oxidation state when discussing heterogeneous catalysis.

Reviewer #3 (Remarks to the Author):

Manuscript Number: NCOMMS-20-19418

Title: The State of Copper under Operando Conditions: Bridging the Gap 3 between Single Nanoparticle Probing and Catalyst-Bed-Averaging

Authors: David Albinsson, Astrid Boje, Sara Nilsson, Christopher Tiburski, Anders Hellman, Henrik Ström, and Christoph Langhammer

The manuscript describes a new catalyst reactor for allowing measurements on the individual nanoparticle, nanoparticle-ensemble, and reactor-distributed-ensemble scales. Perfectly stirred and plug reactors are studied. The reactor is very interesting and appears to offer new and valuable catalyst insights. Unfortunately, insufficient detail is provided regarding the new analytical method and the enabled catalyst observations. It seems that two manuscripts worth of information is being presented as a single manuscript. The work should be divided into an analytical method manuscript and a separate detailed catalyst application manuscript. The catalyst observations are particularly wanting, and may reflect the coauthors' expertise, or it may be simply a result of having too much information for a single manuscript. If expertise is needed, there are very qualified catalyst experts at Chalmers who could be engaged. Given these concerns, a major revision is recommended. Specific supporting comments are provided below.

Positive comments:

- P22 – last sentences are very nice and interesting
- P28L535-P29L540 – very good point

Additional required points:

- The authors give insufficient detail and references to highlight existing analytical methods for probing multiple catalyst scales.
 - E.g., SpaciMS is mentioned, but its strengths and limitations are not adequately addressed. This makes it difficult to fully appreciate the benefits of the new technique. Possibly additional SpaciMS reference could help the authors appreciation. These are numerous and include an introductory chapter of a book dedicated to spatial catalyst characterization.:
 - William P. Partridge and Jae-Soon Choi, "Understanding the Performance of Automotive Catalysts via Spatial Resolution of Reactions inside Honeycomb Monoliths," in Spatially-resolved operando measurements in heterogeneous catalytic reactors, Olaf Deutschmann and Anthony G. Dixon Eds.; Volume 50, Advances in Chemical Engineering Series; Academic Press/Elsevier, Kidlington, United Kingdom (2017) Chapter 1, 1-81, Hardcover ISBN 978-0-12-812589-2.

- The manuscript does not establish or bound the relevance of the reactor
 - Such long residence times relative to many common catalytic reactors
 - Influence of other reactor parameters on relevance
 - Relevance of 120nm x 40nm nanoparticle, compared to typical automotive CO-oxidation catalysts with much smaller PGM nanoparticles. How might the results differ for realistic nanoparticle sizes? What analytical advances are necessary to enable the multi-scale particle/patch measurements on realistic nanoparticle scales?
 - What are the reactor's limitations?
- The manuscript might be better divided into two separate manuscripts. One focusing on the diagnostic and providing a deeper and sufficient dive into the details necessary to prove its ability to measure Cu oxidation state. And a second focusing on applications to understand

catalysis. This second could also go into more sufficient detail; e.g., if/how the hours-long transient timescales are relevant to realistic applications with seconds/minutes transient timescales; referencing catalyst literature (e.g., Goguet at QUBelfast, and Harold at UHouston), investigating catalyst dynamics and hysteresis; etc.

- P11L200-5
 - Why was some more advanced spectral analysis not performed; e.g., chemometrics? This is especially surprising considering how different the different spectra are. The authors should reference literature and discuss this possibility.
 - The two sentences L202-5 don't seem to be sufficiently supported by data
 - The curve in Fig.3b only monotonically decreases for $0 < t < \sim 140$ s.
 - What happens to the sample at $t > 140$ s, and particularly in the transition to plateau at ca. 200s?
 - Does the spectra at $t \sim 140$ look like curve iii in Fig.3a?
 - How does the spectra transition between the curves i and ii in Fig.3a?
 - Have the authors tried to deconvolve the blended spectra, and assign component bands to specific catalyst oxidation states?
 - Is there a direct measure or model of Cu oxidation state showing that it varies linearly with the spectra in Fig.3a?
 - I think the statements could be possible, but need greater evidence and improved care in communication.
- P13-20 – This section is very difficult to read and understand. The structure and language is noticeably different from the first section, and needs better clarity and conciseness.
 - A complete reworking of this section is needed
 - P17L313-5 – The statement and its basis is unclear. And, even with clarification the interpretation seems to assume the proposed diagnostic is a good indicator of Cu oxidation state, which has not been established as mentioned above.
 - P17L320 – is 'max' intended rather than 'main?' The meaning is unclear.
 - P17L324-8 – While the preceding corollary discussion related to the CST reactor is clear, this discussion related to the plug-flow reactor is not.
- P21, Fig.6 – can see some plug-flow behavior in the average and sigma of the five axial rows. This is also shown in Fig.8, but not discussed in either. This obvious and interesting behavior should be discussed. Presumably and not surprisingly, the 'perfectly stirred reactor' is not exactly perfect.
 - In general, the differences in the PSR and Plug reactor are highlighted within the context of their definition and related expectations.
- P24L438 – It looks like Patch 7 has values $\sim 1.8-4.3$, and even Patch 4 looks broader than the stated max.
- P24L439 – which ones are the 'sizable fraction?'
- P24L439-442 – This is not a 'supplementary' statement, and so the data needed to convey the point should be in the main manuscript.
 - This use of such extensive supplementary info could be due to trying to cram two manuscripts worth of information in a single.
 - Similarly, P24L443 makes an interesting point, which should be soundly documented in the main manuscript.
- P24L441-443 – So what does this say about the catalyst and catalyst reactions?
- P24L447 – Contrary to the text, it looks like the terminal patch has individual points scattered over > 30 min. Possibly Patch 8 may have scatter < 30 min.

- P24L452-454 – Of course the macro behavior is the average of many particles, each which has different individual and local-environment characteristics. What's the point? And anyway, what would be the conclusions extracted from the single-particle information? They're not stated, so can't make the stated contrast. The statement thus becomes rather meaningless.
 - This is generally consistent with the lack of substance with respect to catalyst and catalyst reaction insights, and further highlights the author's main interest and expertise in reactor development.
- P25 Fig.8d, terminal Patches – Interesting that Patches 9 and 10 of the plug reactor are more uniform than that of the PSR. This should be discussed, and interpreted in terms of catalyst reactions, and phenomenon origins.
- P26, Section on Catalyst oxidation.. – This whole section is focuses on highlighting the benefits of the new reactor and method rather than unique catalyst observations. The points may be valid, but could be better supported by data/observations in the previous sections.
- P27L512 – not sure the manuscript demonstrated a 'critical' level
- P28L525 – the 'erroneous structure-function relationships' need to be specifically identified
- P28L527-531 – this is known, and is not a revelation resulting from the manuscript

Minor points:

- P6L110 – what is the patch pitch?
- P8L141-2 – QMS measures reactor effluent rather than individual Cu particles as stated. Correct?
- P9 Fig2c – the parameter plotted is 'conversion' and should be called this in the figure caption and related text in Section 1.5 of the Supplemental Material; vs. local relative concentration. It appears to be integral conversion up to a specific axial location in the reactor.
- P11L190-4 – Prior to this sentence, the authors were making the point that by using single-wavelength analysis the measurement speed and resolution could be improved. But then it is stated that the process of analysis using the 'whole visible range' is similar ('Similarly,'). This is confusing, and some clarification might help.
- P13L235 & 241 – need to add 'feed' or 'inlet,' as in '...O₂ feed concentration...'
- P18 Fig5c – use a different symbol for CO₂ vs. the spectrally integrated data.
- P23 Fig7 – why were the specific particle numbers chosen?

Reviewer #1:

The authors David Albinsson et al. used the dark-field microscopy and QMS to study the state of copper under operando conditions. This research bridges the gap between single nanoparticle probing and catalyst-bed-averaging. They provide a great idea to combine the advantages of both dark-field microscopy and QMS to reveal the reaction mechanism during the catalytic reaction in-situ and in real time. The method could make us simultaneously know the relationship between scattering spectra, morphology, products and catalytic activity. A lot of important information has been uncovered, and make us understand the oxidation reaction of CO and copper catalyst. The method in this research is general for many gas phase catalytic reactions and the results are also very important in relative research field. However, there are some problems need to be considered carefully.

We thank the reviewer for the very positive assessment of our work.

Comment 1: The abbreviate QMS in abstract need a full name.

We have reworded to: *"...enables online mass spectroscopic activity measurements."*

Comment 2: How to make a distance 900 nm? A sentence is needed to introduce it.

We interpret the question as the reviewer wondering how it is possible to define a distance as accurately as 900 nm. This is easily possible by the electron beam lithography nanofabrication method that we use, and which is explained in detail in the methods section to which we refer in the introduction of the nanoreactor fabrication sub-section as:

"We have micro-/nanofabricated miniaturized mimics of two types of catalytic reactors into Si/SiO₂ wafers, using the same principles that we have recently introduced,²⁵ and as described in detail in the Methods section, Supplementary information (SI) section 1.1 & Supplementary Fig. S1."

Comment 3: There are some problems in Fig.1. First, the font size is too small to read. Second, the cross section needs a sign of width and height. Third, the b needs to illustrate the structure, which keeps the low pressure in QMS. Fourth, the size of e, f is too small. Actually, these two figures are very important. Fifth, this figure needs to be well organized.

We have addressed the specific points as follows:

1. We have increased the font size to make it more readable.
2. We did not include the arrows for height because it is so small, we believe it is still understandable.
3. We are not sure what the reviewer means here. We only show the estimated pressure caused by the fact that the channel is connected to an external vacuum pump and the UHV chamber of the QMS, which has a base pressure of $1e^{-9}$ mbar.
4. We have increased the size of the four panels in the bottom right.
5. In our opinion the figure as a whole is well-organized and we do not know what specifically needs to be changed in addition to the changes indicated above since no specific suggestion is made by the Reviewer. Hopefully the above changes (1,4) amends this last concern.

Comment 4: Since the Cu nanoparticles are immobilized in the cells of the bed, what's the scattering intensity of these cells as a background.

We are not sure what the reviewer means by "cells". Nevertheless, the scattering background is significantly lower than the scattering from the particles as can be seen in Fig. 1c,d. To account for the background intensity from the camera, we perform background correction as explained in the methods section. The relevant sentence from the manuscript, where this is

discussed, reads as: *“The scattering intensity for particle i (I_i) at every time step was then calculated as $I_i = S_i - I_b$, where S_i is the raw scattering intensity and I_b is the dark current collected by the CCD without any incident light”*

Comment 5: In Figure 2, is it possible to make the grids connect together, and then the gradient of oxygen concentration will be more smooth. This only needs a discussion but not an experiment.

We are not 100 % sure what the Reviewer means with “grids” – we assume s/he refers to the nanoparticle patches. In such case, yes it would be experimentally possible to do this. In this work, however, we have chosen to distribute the nanoparticles in these patches in order to create a scenario where the differences will be clearer and more distinct between patches.

Comment 6: Fig. 4c-e needs a time axis on top of c. In addition, Fig. 4d is not clearly introduced.

The x-axis for c, d & e is the same (O_2 concentration) and thus only explicitly included under panel e. Adding time is not possible in a consistent way since the scaling of the O_2 concentration steps is not linear, and since we include both the increase and decrease in O_2 concentration. To clarify that all panels have the same x-axis, we have added a corresponding comment to the figure caption: *“We note that the x-axis in panels c), d) and e) is the same.”*

We are not sure what specifically is unclear with panel d, therefore we don't really know what to improve. The corresponding caption text reads as: *“d) Measured CO_2 and O_2 outlet concentrations plotted as a function of the nominal inlet O_2 concentration during the O_2 concentration increase (upward triangles) and decrease (downward triangles), respectively.”* We feel that this description is clear.

Comment 7: The color of flow cell in Fig.6a and c may need to be different from that in Fig.6b and d.

We again unfortunately do not really understand what the reviewer means with “flow cell” here. We assume the reviewer is referring to the color coding of the patches inside the reactor schematics in the insets. We have updated them to a different color scheme to hopefully make the figure more readable.

Comment 8: Copper (I) and Copper (II) in the catalyst may play different functions in the catalyst. The authors show some oscillation of the scattering intensity. How do the Copper (I) and Copper (II) contribute to the scattering signal. And how do the Copper (I) and Copper (II) affect the catalytic activity?

This is an interesting and relevant question. From FDTD simulations we know that both these oxides affect the scattering intensity of a Cu particle in a very similar way (since they have similar permittivity in the visible range) and it is therefore unfortunately very hard or even impossible to distinguish them from each other based on the scattering signal from the nanoparticles. Therefore, instead and as a compromise, we utilized ex-situ XPS to identify that Cu_2O is the oxidation state after complete oxidation.

In summary, this paper was written and organized well. This work could have high impact in single particle research field. I am pleased to support the publication in high impact journal like Nature Communications.

We again thank the reviewer for their appreciation of our work.

Reviewer #2 (Remarks to the Author):

This is a novel and carefully executed study that will be of interest to researchers in catalysis and also a wide field of researchers beyond catalysis. The experimental details are provided in great detail as well as the statistical analyses employed.

We thank the reviewer for these kind words.

Comment 1: I only question one conclusion that should be considered in the final version. Heterogeneous catalysis is a surface phenomenon and the manuscript comments about the catalytic relationship with the bulk CuO phase. The bulk phase never directly participates in the catalytic reaction since it is the "surface" of the bulk phase that is involved in the catalytic process. For example, the inactivity of the bulk Cu₂O phase may simply be related to the stabilization of a heavily oxidized surface that does not have any available reduced surface metallic sites for CO adsorption. I think it is important to emphasize the surface oxidation state rather than the bulk oxidation state when discussing heterogeneous catalysis.

This is a relevant comment that we of course can agree with. The reason we mention the "bulk" state is that we observe deactivation of the catalyst along with the observation that oxidation has taken place on a scale that includes the bulk of the particles. In the state where only a thin surface oxide is formed, we find the catalyst still being highly active. Nevertheless, to take the reviewer comment explicitly into account, we have added the following sentence on p.26 of the revised manuscript:

"At the same time, we also note that it ultimately is the surface of the particle on which reaction occurs and thus deactivation most likely occurs already prior to completed oxidation of the bulk of the particle."

We have also addressed the possibility that a combination of sites, as also mentioned by the reviewer, might be responsible for the activity as mentioned in the sentence on row 531 of the original manuscript:

"Furthermore, we observed a highly dynamic oxidation state of the single particles in the regime where the catalyst is found most active, in its most extreme manifestation oscillating between reduced and (partly) oxidized. This suggests that a dynamic mix of metallic Cu and surface oxide constitutes the most active catalyst phase."

Reviewer 3

The manuscript describes a new catalyst reactor for allowing measurements on the individual nanoparticle, nanoparticle-ensemble, and reactor-distributed-ensemble scales. Perfectly stirred and plug reactors are studied. The reactor is very interesting and appears to offer new and valuable catalyst insights. Unfortunately, insufficient detail is provided regarding the new analytical method and the enabled catalyst observations. It seems that two manuscripts worth of information is being presented as a single manuscript. The work should be divided into an analytical method manuscript and a separate detailed catalyst application manuscript. The catalyst observations are particularly wanting, and may reflect the coauthors' expertise, or it may be simply a result of having too much information for a single manuscript. If expertise is needed, there are very qualified catalyst experts at Chalmers who could be engaged. Given these concerns, a major revision is recommended. Specific supporting comments are provided below.

We thank the reviewer for the appreciation of our new reactor platform. At the same time, we are a bit confused about the reviewer both stating that we provide insufficient detail, while at the same time saying that we provide too much information for one manuscript and rather should divide it into two. To this end, as we detail below, we strongly disagree with this assessment and firmly believe that our results and analysis are to be published as one single piece of work. Regarding the lack of information, we hope that the additional explanations added to the revised manuscript can help clarify the issues raised. Furthermore, we want to highlight that reviewer # 2 states that "the experimental details are provided in great detail as well as the statistical analyses employed". When it comes to the comment about the (lack of) catalysis expertise of the authors and the availability of such expertise at Chalmers, we note that Prof. Anders Hellman, who is one of our co-authors, indeed is affiliated the Chalmers' Competence Center for Catalysis and thus represents this group of "qualified experts".

Positive comments:

- P22 – last sentences are very nice and interesting
- P28L535-P29L540 – very good point

We thank the reviewer for these positive comments.

Negative comments:

Comment 1: The authors give insufficient detail and references to highlight existing analytical methods for probing multiple catalyst scales. E.g., SpaciMS is mentioned, but its strengths and limitations are not adequately addressed. This makes it difficult to fully appreciate the benefits of the new technique. Possibly additional SpaciMS reference could help the authors appreciation. These are numerous and include an introductory chapter of a book dedicated to spatial catalyst characterization: William P. Partridge and Jae-Soon Choi, "Understanding the Performance of Automotive Catalysts via Spatial Resolution of Reactions inside Honeycomb Monoliths," in Spatially-resolved operando measurements in heterogeneous catalytic reactors, Olaf Deutschmann and Anthony G. Dixon Eds.; Volume 50, Advances in Chemical Engineering Series; Academic Press/Elsevier, Kidlington, United Kingdom (2017) Chapter 1, 1-81, Hardcover ISBN 978-0-12-812589-2.

The reviewer is correct that we do not go through all existing techniques that enable the probing of multiple catalyst scales in detail, since we do not think it is within the scope of this work. When it comes to the Spaci-MS specifically, we did not explicitly discuss it because it operates at length scales that are 100 - 1000 times larger than our reactor/catalyst bed. Therefore, we felt that a detailed direct comparison was not motivated since the two methods are very different. However, as requested by the reviewer, we have added the following sentences to the revised introduction:

“For example, the thin 150 mm capillary used in Spaci-MS² enables spatiotemporal characterization of reactant gradients with a spatial resolution of ca. 300 μm inside catalyst monoliths. However, due to the size of the capillary, it is not possible to directly measure concentration gradients inside the porous catalyst material itself, since the pores typically are more than 1000 times smaller.”

Comment 2: The manuscript does not establish or bound the relevance of the reactor:

- Such long residence times relative to many common catalytic reactors

We did not specify the residence time of our reactors in the original version of the manuscript anywhere. Thus, we thank the reviewer for making us aware of this. For reference we note that the residence time is approximately 6 ms in the PFR-like and 60 ms in the CSTR-like reactor, respectively. We do not consider these residence times to be particularly long for a research reactor. For clarity, we have now added these numbers to the revised manuscript as:

“These designs resulted in reactant residence times of approximately 6 ms and 60 ms in the plug-flow-type and well-mixed reactor, respectively.”

- Influence of other reactor parameters on relevance

This is an interesting question, which at the same time raises the issue of what “relevance” means. It is not 100% clear from the Reviewer’s comment but we presume it refers to a “real” catalyst or reactor. Hence, it refers to the structure gap and the long-standing question of the relevance of model studies and model catalysts. To this end, there are probably as many opinions as there are scientists working in the field and we think we can all agree that a model-catalyst study never can be 100% directly translated to the behavior of a “real” catalyst because too many parameters will be different. At the same time, we also think that everyone can agree on the fact that model studies have proven instrumental for advancing our understanding of catalytic processes since they enable a level of control/simplification that make it possible to study/highlight specific aspects of a catalyst process/system. In light of this, based on our current knowledge, the reactor platform discussed in our work provides a more well-defined and controllable environment with fewer unknown parameters compared to a conventional reactor/catalyst, and it was specifically designed to highlight the effects discussed in the manuscript. However, since it is difficult, if not impossible, to probe the state of a single catalyst nanoparticle inside a conventional reactor, there are, to the best of our knowledge, no reference measurements made in a conventional system to directly compare our results to. However, when comparing our results to a study on a much larger model catalyst (Bu et.al. ACS Catal 6 , 2016 – Ref 28 in manuscript), the overall behaviors are quite similar, meaning that our observations are not a direct consequence of the particular reactor design. To highlight this point, we have added the following sentence and reference on page 13-14 of the revised manuscript:

“This is similar to the observations made in a study performed on a much larger model catalyst,²⁸ indicating that our results are not only related to the specific reactor geometry.”

- Relevance of 120 nm x 40 nm nanoparticle, compared to typical automotive CO-oxidation catalysts with much smaller PGM nanoparticles. How might the results differ for realistic nanoparticle sizes? What analytical advances are necessary to enable the multi-scale particle/patch measurements on realistic nanoparticle scales?

This comment directly relates to the question above and the relevance of model studies in general. Hence, we will not repeat the main points of our response given above. However, it is of course clear that one of the main directions of future efforts in this area must be to ultimately enable the same type of study on nanoparticles in the sub-10 nm size range. If we are to speculate about how results for such particles may differ from the ones observed for the particles at hand in the present study, we believe not so much, and for some observations not at all. For example, when it comes to the effects of reactor design and conversion, we highlight that in one of the models used to reproduce the experimental data the particles are *not* explicitly included in terms of size. Therefore, it is very reasonable to assume that the same behavior would be observed for PGM nanoparticles. When it comes to the single-particle specific behaviors, we would expect strong particle-specific effects also in the sub-10 nm size range since particle heterogeneity in terms of morphology, defects and size prevails. Energetics of oxide formation/reduction are likely to vary (and may become size dependent) and thus the exact conditions at which specific phases are formed may vary as well. To explicitly discuss the limitations of our method in its current state, we have added the following sentences to the conclusions section:

“The method presented here is currently limited to particles with a size > 50 nm in at least one dimension in order to ensure a detectable optical contrast from the single catalyst nanoparticles. Since this is one order of magnitude larger than technically used catalyst nanoparticles, which typically are in the sub-10 nm size range, it is of interest to implement strategies like indirect nanoplasmonic sensing to enable the study of catalyst nanoparticles in this size regime in the future.”⁴⁷⁻⁴⁹

- What are the reactor’s limitations?

Since the reactor platform is purposely designed to be modular, and relies on nanofabrication methods that essentially enable almost “any” design of the fluidic structures, it is possible to design widely different reactor geometries tailored to address a specific question or catalyst system. To, nevertheless, discuss some technical limitations of the current design, we note that it is difficult to study the catalyst under temperature-transient conditions since thermal expansion of the chip will lead to movement of the catalyst bed, which in turn will move it out of the field of view of the microscope and/or out of focus. In other words, currently only experiments under constant temperature conditions are possible. However, this limitation may be resolved by installing an optical feedback loop together with micro- and nanopositioners that correct for sample movement and focus continuously. A second limitation of the system is the operating temperature, which currently is max. 450°C due to the design of the resistive heater. The temperature limitation is stated in the methods section in the sentence:

“On the back of the chip, a Pt thin film heater and resistive thermometer is placed and the on-chip temperature is controlled using a PID (Lakeshore 335) to enable precise temperature control of the reaction zone of the chip in the range from 20°C to 450°C.”

Comment 3: The manuscript might be better divided into two separate manuscripts. One focusing on the diagnostic and providing a deeper and sufficient dive into the details necessary to prove its ability to measure Cu oxidation state. And a second focusing on applications to understand catalysis. This second could also go into more sufficient detail; e.g., if/how the hours-long transient timescales are relevant to realistic applications with seconds/minutes transient timescales; referencing catalyst literature (e.g., Goguet at QUBelfast, and Harold at UHouston), investigating catalyst dynamics and hysteresis; etc.

Here we strongly disagree with the reviewer for the following reasons:

- 1) Any results related to “understanding catalysis” are tightly related to the experimental approach used, since we have specifically designed the two reactor types and catalyst beds for this purpose. Hence, presenting them separately makes no sense since any reader will need both types of information in the same body of work.
- 2) As discussed in more detail also further below in response to related comments, we disagree with the necessity for a separate deep-dive into the ability of probing the oxidation state of Cu nanoparticles using plasmonic resonance based optical readout. This approach is very well established, as proven by multiple publications both by ourselves and other groups (Rice, Paterson and Stoykovich, 2015; Susman, Vaskevich and Rubinstein, 2016; Susman *et al.*, 2017; Albinsson *et al.*, 2019; Nilsson *et al.*, 2019). Therefore, a separate study of this kind would not produce anything new and therefore we are also very confident about our interpretation of the optical data. To make this clear, we have added the following sentence and references to the revised manuscript:

“The optical response from Cu nanoparticles has previously been used to study oxidation of both particle ensembles⁴¹⁻⁴³ and individual nanoparticles³⁹ and to study the state of Cu nanoparticle catalysts.^{25,28}”

- 3) The reviewers’ second concern related to long transient time scales is most likely a misunderstanding. The residence times in our reactors are on the scale of tens of milliseconds and the concentration gradients presented are steady state gradients. The long time scales we think the reviewer is referring to are related to the steps we stay in at each gas mixture. In a real catalyst we assume that steady state conversions are of interest and what we show is that it is difficult to keep a Cu catalyst in an active state without limiting the inlet oxygen concentration significantly.

Comment 4: P11L200-5

- Why was some more advanced spectral analysis not performed; e.g., chemometrics? This is especially surprising considering how different the different spectra are. The authors should reference literature and discuss this possibility.

We agree with the reviewer that more advanced chemometrics could be very useful for analyzing spectral data related to the plasmonic response of the Cu nanoparticle system. This is something we are actively pursuing in a parallel project where the oxidation of Cu in the

absence of CO is studied. For example, a form of chemometrics was employed to understand different spectral features during Cu oxidation in (Albinsson *et al.*, 2019). Similarly, in a later study of single Cu nanoparticles we used a form of chemometrics to identify specific spectral characteristics related to the formation of a void due to the nanoscale Kirkendall effect. (Nilsson *et al.*, 2019, ref 39)

In the study we present here *during CO oxidation reaction*, we do not collect the whole spectra for all particles, instead we rely on the integrated intensity by taking images. This makes it possible to study on the order of 1000 particles instead of ca. 20 in parallel, which is one of the main points of the present work. The price for being able to study this many single particles simultaneously is of course the loss of spectral information and therefore we are not able to perform advanced chemometrics at the same time. Looking ahead we are envisioning the use of more advanced hyperspectral imaging techniques to be able to combine imaging and spectral information and this could probably lead to even more interesting findings related to the exact state of the individual particles.

- The two sentences L202-5 don't seem to be sufficiently supported by data
 - The curve in Fig.3b only monotonically decreases for $0 < t < \sim 140$ s.

We realize based on these two comments, as well as the general comment above, that we may have (wrongly) assumed that it is well established enough that oxidation of Cu nanoparticles is directly correlated to a change in their optical plasmonic signature – and thus scattering intensity – and that we therefore may not have made this point clear enough in the manuscript by the appropriate references (also to not have an excessively long reference list). In addition to the corresponding changes introduced in the context of the general comment above:

“The optical response from Cu nanoparticles has previously been used to study oxidation of both particle ensembles^{41–43} and individual nanoparticles³⁹ and to study the state of Cu nanoparticle catalysts.^{25,28}”

we have also modified the sentence L202-5 to:

“Clearly, this integrated scattering intensity exhibits an almost monotonic decrease over time during the exposure to oxygen (gray background) until a new equilibrium state is reached at the end of the oxidation process.”

- What happens to the sample at $t > 140$ s, and particularly in the transition to plateau at ca. 200s?

In general, the monotonic decrease in total scattering intensity stems from the decrease of the metallic LSPR peak that both decreases and red-shifts until it completely disappears. Then, at the end of the oxidation process, there is a slight increase in scattering intensity in the wavelength range (< 700 nm) that takes place in a regime where no metallic signature is present anymore, which means it is not related to the oxidation of metallic Cu. It is this increase that gives rise to small minimum observed in Fig. 3b before the plateau is reached at $t \sim 200$. Speculatively, this effect can be attributed to a structural change of the particle in the oxide state or to the formation of CuO, as suggested by Bu, et al. ACS Catal 2016. However, the Bu et al. measurements were done in transmission mode on a nanoparticle ensemble and can therefore not be directly translated to scattering from a single nanoparticle. Therefore, we

hesitate to propose this as the reason. To include the above discussion in the main text, we have added the following sentences in the revised manuscript:

“The slight minimum at $t \sim 150$ s is the consequence of the slight increase in scattering intensity in the 500 – 700 nm spectral range observed during the final phase of the oxidation. It can tentatively be explained either by a structural change of the particle in the oxide phase or a change in the oxidation state of the oxide as proposed by Bu et al.²⁸.”

As a second point, we want to emphasize here that this effect is only seen during oxidation in pure O₂. It is absent during the CO oxidation experiments (see Fig. 4 & 5). This means that it (i) is not important for the interpretation of the catalysis results and (ii) if CuO formation would be the origin, it would corroborate our XPS data that confirm the absence of the CuO phase after oxidation at CO oxidation reaction conditions.

- Does the spectra at $t \sim 140$ look like curve iii in Fig.3a?

We have added additional spectra to Figure 3a (see below) to also explicitly show the spectrum at 140 s (now marked “iii” in the new figure). Evidently the spectra at $t \sim 140$ and $t \sim 200$ are both flat and the plasmonic – and thus metallic – signature of the particle has completely disappeared. The only difference is a slightly lower scattering intensity in the short wavelength range, where a small increase in scattering intensity takes place from $t \sim 140$ to $t \sim 200$, as discussed above.

- How does the spectra transition between the curves i and ii in Fig.3a?

It decreases in intensity and shifts to the right and exhibits a peak split, in very good agreement with our previous work on the oxidation of single Cu nanoparticles in pure O₂ (Nilsson *et al.*, 2019). To illustrate this more clearly, we included more spectra in between i and ii in the updated Figure 3a. and added the following sentence to the corresponding discussion in the main text:

“The spectral evolution is in very good agreement with a corresponding study of single Cu nanoparticle oxidation by Nilsson et al.³⁹”

- Have the authors tried to deconvolve the blended spectra, and assign component bands to specific catalyst oxidation states?

Yes, we have tried this. However, assigning specific bands is difficult since each nanoparticle is slightly different in its morphology, leading to varying locations of the initial band. Also, the formation of Kirkendall voids at different spatial locations within a particle during oxidation can give rise to significantly different “band locations” (Nilsson *et al.*, 2019). Furthermore, where such void formation starts spatially in a nanoparticle depends strongly on the specific morphology of that particle (e.g. defect density, grain boundaries, etc.), as our ongoing unpublished study of this specific topic shows.

- Is there a direct measure or model of Cu oxidation state showing that it varies linearly with the spectra in Fig.3a?

To show this is essentially the purpose of the data we present in Fig. 3b, with the intention to show that although the spectral features can be quite dramatic, the intensity decreases almost monotonically. The same observation has been made previously in a separate study of the oxidation of single Cu nanoparticles from our group (Nilsson *et al.*, 2019), and therefore we feel confident that it is a safe assumption to make.

- I think the statements could be possible, but need greater evidence and improved care in communication.

We hope that the above discussion and corresponding additions and changes to the manuscript and Figure 3 have convinced the reviewer that our statements indeed are sound and correct.

Comment 5: P13-20 – This section is very difficult to read and understand. The structure and language is noticeably different from the first section, and needs better clarity and conciseness.

We have tried to further polish the text where we found it necessary. All changes are indicated by the tracked changes in the updated manuscript.

- P17L313-5 – The statement and its basis is unclear. And, even with clarification the interpretation seems to assume the proposed diagnostic is a good indicator of Cu oxidation state, which has not been established as mentioned above.

As we have discussed in detail in response to related comments above, it is indeed well established that the optical response measured is a direct indicator of the Cu oxidation state. We hope that with the additional references and text provided, it is clear that this statement is correct .

- P17L320 – is ‘max’ intended rather than ‘main?’ The meaning is unclear.

Main is correct. However, to clarify our message we modified the sentence, which now reads:

“Focusing first on the well-mixed reactor, where the local concentration at the particles is considered the same throughout the reactor (cf. Fig. 2a), we note that the main oxidation event (steepest decline in intensity) is observed at an inlet O_2 concentration of 0.3 %.”

- 17L324-8 – While the preceding corollary discussion related to the CST reactor is clear, this discussion related to the plug-flow reactor is not.

We have modified the section - it now read as:

“Turning to the plug-flow reactor, we observe bulk oxidation of the first patch at an inlet O_2 concentration of 0.12 %. Translated to the local concentration, based on the simulated concentration profiles presented in Fig. 2b, the first patch oxidizes at an O_2 concentration of 0.074 %. In other words, we observe essentially the same local O_2 concentration required for bulk oxidation in both reactor types”

Comment 6: P21, Fig.6 – can see some plug-flow behavior in the average and sigma of the five axial rows. This is also shown in Fig.8, but not discussed in either. This obvious and interesting behavior should be discussed. Presumably and not surprisingly, the ‘perfectly stirred reactor’ is not exactly perfect.

This is a relevant observation and we agree on all points. The reason we choose not to discuss this effect explicitly for the well-mixed reactor in the original manuscript was that this plug-flow-type behavior is more pronounced in the PFR, which is designed to maximize it. However, there is absolutely no problem to add a comment on its milder appearance also in the well-mixed reactor, as the reviewer requests. Since we believe it is most clearly illustrated in Fig. 8, we have added the following statement in the discussion related to that figure:

“We also observe a slight position dependence on the τ_{ox} where particles placed further up-stream oxidize prior to particles further downstream. This indicates that we have a slight PFR-like behavior also in the well-mixed reactor, as also predicted by our model (Fig. 2a,c).”

In general, the differences in the PSR and Plug reactor are highlighted within the context of their definition and related expectations.

Comment 7: P24L438 – It looks like Patch 7 has values ~1.8-4.3, and even Patch 4 looks broader than the stated max.

This is correct, but we treat the individual particles that oxidize very early as outliers in our analysis. The range we state in the text is based on the box plots with whiskers presented under the histograms. We have modified a sentence to specifically emphasize this. It now reads:

“We also see that the distribution is quite broad, and broadest for patches closest to the inlet. In the most extreme case, within one patch, individual particles exhibit τ_{ox} that range from 2.8 – 4.4 hours (based on the whiskers in the box plots to not overemphasize the few even more extremely spread particles).”

Comment 8: P24L439 – which ones are the ‘sizable fraction?’

The fraction is shown in the supplementary figure S8. To clarify we have added a number to this statement which now reads:

*“Furthermore, we find that a sizable fraction of single particles is considered oxidized (i) before the QMS signal indicates a loss in catalytic activity (ca. 90%, **Fig. 8a,b** & Supplementary Fig. S8a) and (ii) before the average optical signal indicates bulk oxidation (ca. 50%, **Fig. 8a,b** & Supplementary Fig. S8b).”*

Comment 9: P24L439-442 – This is not a ‘supplementary’ statement, and so the data needed to convey the point should be in the main manuscript.

We have taken the reviewers opinion into consideration but in the end do not agree that these figures present enough new information to be needed in the main text. Essentially, it is the same data as presented in Fig. 8, but the oxidation event is presented as a cumulative sum instead of histograms of their exact times. Therefore, we keep them in the supplementary information.

- This use of such extensive supplementary info could be due to trying to cram two manuscripts worth of information in a single.

It is of course up for debate and in the end a matter of taste/understanding what the purpose of a supplementary information is. We see it as a tool to supply the interested expert reader with in-depth information that provides the basis for what is presented in the main text but that not important enough to be explicitly shown there. In this way, a manuscript becomes more understandable and accessible to readers across disciplines, while at the same time also providing the necessary details for the topical experts. In our opinion, for a manuscript to be published in a cross-disciplinary journal with a broad scope like Nature Communications, this way of writing is essential - which is why we have done so. It was by no means an attempt to cram two works into one. As we have expressed in reply to comment 3, we do not believe this work would be better presented as two separate papers. As further reply to this point, we also refer to our above statement related to the reviewer suggestion to split our work into two.

- Similarly, P24L443 makes an interesting point, which should be soundly documented in the main manuscript.

The specific statement is discussed in the reply to the comment below.

In line with our view of the purpose of supplementary information outlined above, we believe that the point raised by the reviewer is well placed in the SI, where it is adequately discussed.

Comment 10: P24L441-443 – So what does this say about the catalyst and catalyst reactions?

This is an interesting question and we attribute the more rapid reduction to the fact that the amount of reducing agent (CO) is not rate limiting. i.e. there is plenty of CO available to reduce CO, and the slight decrease in CO concentration inside the reactor will not affect particles placed downstream. Actually, we address this point explicitly already in the main

text of the original manuscript on line 337-340, and we reproduce the sentence below for convenience:

“This is in good agreement with what we observed in the well-mixed reactor and thus with the overall scenario since the rate of Cu oxide reduction is not limited by the availability of CO as it is present in excess.”

We acknowledge that this information might need to be repeated so we updated the sentence that now reads:

*“For the reversed process of particle reduction when decreasing the O₂ concentration in the feed we find very narrow distributions in all patches which peak simultaneously, irrespective of position in the reactor (Supplementary **Fig. S9a**) that can be explained by the excess of the reducing agent CO.”*

Comment 11: P24L447 – Contrary to the text, it looks like the terminal patch has individual points scattered over >30min. Possibly Patch 8 may have scatter <30min.

Similar to the answer above, we use the box plot with whiskers to make this analysis. In this way we ignore outliers in the data, since we find it to be a more statistically relevant way of analyzing our data. We have, for clarity, added this information in the sentence which now reads:

*“For the most extreme case of the most upstream patch, τ_{ox} of the individual particles is spread out over almost 3 hours, whereas within the array closest to the outlet all particles bulk oxidize within only 25 minutes (based on whiskers in **Fig. 8d**).”*

Comment 12: P24L452-454 – Of course the macro behavior is the average of many particles, each which has different individual and local-environment characteristics. What’s the point? And anyway, what would be the conclusions extracted from the single-particle information? They’re not stated, so can’t make the stated contrast. The statement thus becomes rather meaningless. This is generally consistent with the lack of substance with respect to catalyst and catalyst reaction insights, and further highlights the author’s main interest and expertise in reactor development.

We find it quite surprising that the statement invoked is found meaningless – it reads: *“Consequently, utilizing the information extracted from a single particle leads to significantly different conclusions than the patch averaged response with respect to catalyst state and active phase.”*. We feel that the detailed preceding discussion in the manuscript provides ample support for why this claim is important. To make it clear here, the key message is the fact that it is important to study both the ensemble AND the single particles to fully understand the catalyst. For example, as the reviewer also has mentioned above, in Fig. 8 there are numerous “outlier particles”. If we were to only study a single nanoparticle, there is a risk that we measure one of these outliers. And if we were to utilize that information to make predictions of the state of the whole catalyst bed this could lead to an incorrect understanding of the structure function relationship. Furthermore, it is not correct to state that we do not mention the value of the single particle information. Specifically, as an example, we write in the sentence preceding P24L452-454:

”Comparing the τ_{ox} distribution of the individual particles in each patch (Fig. 8d) with the averaged scattering response from each patch (Fig. 8c) reveals that the steplike and fluctuating average response is a consequence of the individual particles in the patch oxidizing at very different and particle-specific times rather than a collective behavior of all particles within the patch (see SI section 1.7 for detailed discussion).”

Comment 13: P25 Fig.8d, terminal Patches – Interesting that Patches 9 and 10 of the plug reactor are more uniform than that of the PSR. This should be discussed, and interpreted in terms of catalyst reactions, and phenomenon origins.

Indeed, they are. We explain this by the fact that the local oxygen concentration during the main oxidation event is higher for particles placed further down-stream, as discussed in detail in SI section 1.7. We have added additional discussion related to this into the main text:

“The narrower τ_{ox} distribution observed for particles oxidized later can be rationalized by remembering that the O_2 concentration is increased during the experiment and that fewer particles are actively consuming O_2 due to deactivation. Both these effects contribute to a larger O_2 excess available for particles placed further downstream and consequentially a faster (less mass transport limited) Cu oxidation.”

Comment 14: P26, Section on Catalyst oxidation. – This whole section is focuses on highlighting the benefits of the new reactor and method rather than unique catalyst observations. The points may be valid, but could be better supported by data/observations in the previous sections.

This section begins by briefly summarizing the disagreement present in the literature followed by what we have observed in this context. We highlight that our findings indicate that a partially oxidized particle is most active and we further discuss the possibility of a dynamic state during the active phase. This is not related to our reactor, but solely focuses on the catalyst itself. We also highlight the importance of operando techniques capable of studying many (preferably all) catalyst nanoparticles in parallel. This is a general comment which applies to catalyst characterization.

Comment 15: P27L512 – not sure the manuscript demonstrated a ‘critical’ level

In this sentence we refer to the ability to characterize catalyst beds under operation with high spatial resolution as well as a large field of view *in general*. We are not referring to our method specifically, as the reviewer claims. However, critical might have been a too powerful word in any case, thus we have changed it to “an important”.

Comment 16: P28L525 – the ‘erroneous structure-function relationships’ need to be specifically identified

We want to highlight that we write “*potentially* erroneous” – where *potentially* is the key word – since we do not know for certain which investigations have identified the correct structure function relations. Our point here is that the measurement method might influence the conclusions in many ways.

Comment 17: P28L527-531 – this is known, and is not a revelation resulting from the manuscript

We find it surprising that the Reviewer above (in the “Positive Comments”) states our sentence about particle differences is very relevant and interesting, and now here finds it obvious. This makes it a bit difficult to respond. In any case, regardless if this should be a well-known fact, there are studies published where this fact is not taken into account. Furthermore, to our knowledge, there are not many (if any – we are at least not aware of any) studies that experimentally illustrate these effects at the length scales of our work and with single nanoparticle resolution. Thus we are convinced that it is a finding worth highlighting in this sentence – hopefully bringing the “obvious” explicitly to the attention of more scientists in the relevant fields.

Minor points:

- P6L110 – what is the patch pitch?

The patches are placed 20 μm apart. Added this to the text on line 108-109 which now reads:

” ... “patches” containing 100 nanoparticles each placed 20 μm apart, as detailed in Fig. 1.”

- P8L141-2 – QMS measures reactor effluent rather than individual Cu particles as stated. Correct?

This is correct. To clarify we modified the sentence to say catalyst bed instead of nanoparticles. It now reads:

“This design enables online QMS measurements of the Cu catalyst bed...”

- P9 Fig2c – the parameter plotted is ‘conversion’ and should be called this in the figure caption and related text in Section 1.5 of the Supplemental Material; vs. local relative concentration. It appears to be integral conversion up to a specific axial location in the reactor.

What we plot is not conversion, rather is it the local concentration of O_2 relative to the incident concentration (C/C_0). Conversion would have been $1-C/C_0$ and would thus essentially flip the 1 and 0 positions on the y-axis. We believe that both representations can be used and convey the information correctly.

- P11L190-4 – Prior to this sentence, the authors were making the point that by using single- wavelength analysis the measurement speed and resolution could be improved. But then it is stated that the process of analysis using the ‘whole visible range’ is similar (‘Similarly,’). This is confusing, and some clarification might help.

We write that speed can be improved by using a single wavelength measurement since many hyperspectral techniques relies on taking images and many wavelengths on after another. The “similarity” is that we rely on a single image, but we do not measure a narrow wavelength span but instead use a white light (broad band).

We have tried to clarify this by modifying the sentence, which now reads:

“Similarly, in the present study, we have relied on white light imaging to enable multiplexed readout of potentially thousands of single nanoparticles simultaneously. In this solution,

which we call multiplexed single particle plasmonic nanoimaging, optical contrast is generated by measuring changes in the scattering intensity in the whole visible range from each of the single nanoparticles.”

- P13L235 & 241 – need to add ‘feed’ or ‘inlet,’ as in ‘...O₂ feed concentration...’

Indeed, this has been added.

- P18 Fig5c – use a different symbol for CO₂ vs. the spectrally integrated data.

These symbols were there to illustrate that the O₂ concentration was increasing. Since we have not included the decrease (as we did in Fig. 4) we have removed the symbols completely in the updated figure, and hope this makes the figure clearer.

- P23 Fig7 – why were the specific particle numbers chosen?

We selected them based on the fact that they show a good representation of different interesting behaviors. We added “selected” in the description to emphasize that this was not a random subset.

References

Albinsson, D. *et al.* (2019) ‘Heterodimers for in Situ Plasmonic Spectroscopy: Cu Nanoparticle Oxidation Kinetics, Kirkendall Effect, and Compensation in the Arrhenius Parameters’, *The Journal of Physical Chemistry C*. American Chemical Society, 123(10), pp. 6284–6293. doi: 10.1021/acs.jpcc.9b00323.

Albinsson, D. *et al.* (2020) ‘Operando detection of single nanoparticle activity dynamics inside a model pore catalyst material’, *Science Advances*, 6(25), p. eaba7678. doi: 10.1126/sciadv.aba7678.

Bu, Y., Niemantsverdriet, J. W. H. and Fredriksson, H. O. A. (2016) ‘Cu Model Catalyst Dynamics and CO Oxidation Kinetics Studied by Simultaneous in Situ UV-Vis and Mass Spectroscopy’, *ACS Catalysis*, 6(5), pp. 2867–2876. doi: 10.1021/acscatal.5b02861.

Morgan, K. *et al.* (2016) ‘Evolution and Enabling Capabilities of Spatially Resolved Techniques for the Characterization of Heterogeneously Catalyzed Reactions’, *ACS Catalysis*. American Chemical Society, 6(2), pp. 1356–1381. doi: 10.1021/acscatal.5b02602.

Nilsson, S. *et al.* (2019) ‘Resolving single Cu nanoparticle oxidation and Kirkendall void formation with in situ plasmonic nanospectroscopy and electrodynamic simulations’, *Nanoscale*. The Royal Society of Chemistry, 11(43), pp. 20725–20733. doi: 10.1039/C9NR07681F.

Rice, K. P., Paterson, A. S. and Stoykovich, M. P. (2015) ‘Nanoscale kirkendall effect and oxidation kinetics in copper nanocrystals characterized by real-time, in situ optical spectroscopy’, *Particle and Particle Systems Characterization*, 32(3), pp. 373–380. doi: 10.1002/ppsc.201400155.

Susman, M. D. *et al.* (2017) ‘Real-time plasmon spectroscopy study of the solid-state oxidation and Kirkendall void formation in copper nanoparticles’, *Nanoscale*. Royal Society

of Chemistry, 9(34), pp. 12573–12589. doi: 10.1039/C7NR04256F.

Susman, M. D., Vaskevich, A. and Rubinstein, I. (2016) ‘A General Kinetic-Optical Model for Solid-State Reactions Involving the Nano Kirkendall Effect. The Case of Copper Nanoparticle Oxidation’, *The Journal of Physical Chemistry C*. American Chemical Society, 120(29), pp. 16140–16152. doi: 10.1021/acs.jpcc.6b00137.

REVIEWER COMMENTS

Reviewer #1 (Remarks to the Author):

The manuscript has been revised accordingly. I suggest to be published as current state.

Reviewer #3 (Remarks to the Author):

Comments to the Authors

Manuscript Number: NCOMMS-20-19418; Revision 1

Title: The State of Copper under Operando Conditions: Bridging the Gap 3 between Single Nanoparticle Probing and Catalyst-Bed-Averaging

Authors: David Albinsson, Astrid Boje, Sara Nilsson, Christopher Tiburski, Anders Hellman, Henrik Ström, and Christoph Langhammer

Thank you to the manuscript authors for pointing out this reviewer's overlooking Anders Hellman's connection with KCK, and compliments for appropriately noting this important link in the revised manuscripts. For some points, there seems to be some communication barrier between this reviewer's intended points and the authors' understanding. For other points there is clear difference of opinion, which is the authors' prerogative. Indeed, the reactor is very nice and has great potential for unique catalyst studies. Unfortunately, the lack of instrument detail and superficial and biased discussion of the catalyst application, make it neither the great instrument nor catalyst manuscript that it could be. This reviewer's intention was to motivate the authors to seize that potential. From the responses it seems that the authors are intentionally choosing something different. Despite my frustrations with the authors' approach, I compliment them on their reactor development and look forward to seeing their catalyst applications.

Regarding some specific comments:

- "...we are a bit confused about the reviewer both stating that we provide insufficient detail, while at the same time saying that we provide too much information for one manuscript and rather should divide it into two."
 - o The manuscript covers instrumentation and application subtopics. Less than sufficient detail is given to both subtopics. The authors seem to acknowledge this in some responses. For instance, in justifying their superficial application analysis as due to the need to easily summarize their intended point without greater detail.
- Regarding catalyst scale spatial resolution, the point was to establish the relevance, benefits and complimentary nature of the new reactor/instrument with respect to these more established techniques. The point was not to reference SpaciMS specifically. There are many similar, in terms of

spatial resolution but individually unique, techniques that have been developed independently (Partridge, Horn, Epling, Goguet, Harold, etc.) across the catalyst community, and these are discussed in the provided references. Again, it is this reviewer's opinion that the authors shortchange the quality of their manuscript by not orienting their contribution with respect to existing technologies and literature.

- It is not meaningless that the behavior of individual catalyst sites differs from the ensemble, but it's not surprising or unknown. What's meaningless is to state this obvious fact without discussing relevance, origin, pathways to advanced performance enabled by the new methodology, etc. It is very nice that this new reactor provides a path for studying variations in individual site nature, and how these influence the bulk performance.

Point-to-point response

Reviewer #1 (Remarks to the Author):

The manuscript has been revised accordingly. I suggest to be published as current state.

Our reply: We thank the reviewer for his/her time and effort and for suggesting our work for publication.

Reviewer #3 (Remarks to the Author):

Thank you to the manuscript authors for pointing out this reviewer's overlooking Anders Hellman's connection with KCK, and compliments for appropriately noting this important link in the revised manuscripts. For some points, there seems to be some communication barrier between this reviewer's intended points and the authors' understanding. For other points there is clear difference of opinion, which is the authors' prerogative. Indeed, the reactor is very nice and has great potential for unique catalyst studies. Unfortunately, the lack of instrument detail and superficial and biased discussion of the catalyst application, make it neither the great instrument nor catalyst manuscript that it could be. This reviewer's intention was to motivate the authors to seize that potential. From the responses it seems that the authors are intentionally choosing something different. Despite my frustrations with the authors' approach, I compliment them on their reactor development and look forward to seeing their catalyst applications.

Our reply: We agree with the reviewer that there are some communication barriers, as we also have highlighted in our first response specifically, when we were not sure how to exactly understand the feedback provided by the reviewer. However, we had tried our best to respond in an appropriate and detailed manner.

As the next point, the reviewer mentions a “lack of instrumental detail” in the manuscript. Relating to the aforementioned communication barriers, also here, since no specific details are given, it is hard for us to understand what exactly the reviewer is missing. In our opinion, our device and instrument are described in great detail in the manuscript and its supporting information, as well as in reference 25. Therefore, and since no specific lacking points are mentioned, we think that the provided information is sufficient to understand, as well as reproduce, our instrument in general and the nanoreactor in particular.

Further, the reviewer states that we provide a “superficial and biased” discussion of the catalyst application, but s/he is not providing any explicit examples for what is meant neither with “biased” nor with “superficial”. When it comes to the “superficial” point, we think that a manuscript that is 45 pages long and accompanied by more than 20 pages supporting information can hardly be a work that is intentionally superficial. Rather, we have done our best to provide details, control experiments and extensive theoretical simulations and calculations as the basis for a detailed discussion of our results. Therefore, we conclude that the notion that our discussion is “superficial” most likely is the consequence of the different views on what this paper is about between the reviewer and the authors. When it comes to “biased”, it is unclear to us what kind of bias the reviewer has in mind. Naturally, we are

presenting this work from our perspective and maybe this is the bias that s/he refers to. In this sense, any work will be biased and different scientists have different views and preferences for how a specific work should be presented.

Finally, we also (again) would like to explain why we strongly believe that splitting our work into two separate “instrument” and “catalysis” manuscripts would not work out very well. Starting with the “instrument” manuscript, if we remove the catalysis part, what would remain – and most importantly, how would we demonstrate convincingly that our instrument is really useful? Turning to the “catalysis” manuscript - how would we be able to understandably explain and interpret the results without including first a detailed discussion of the used instrument/nanoreactor? It would essentially be impossible because the obtained catalysis results are a direct consequence of/are enabled by the new instrument/nanoreactor we use. Therefore, for us, the best way to publish this work, where this type of nanoreactor and readout is used for the first time, is to combine the two aspects in one and the same work, as we have done.

Regarding some specific comments:

- *“...we are a bit confused about the reviewer both stating that we provide insufficient detail, while at the same time saying that we provide too much information for one manuscript and rather should divide it into two.”*

The manuscript covers instrumentation and application subtopics. Less than sufficient detail is given to both subtopics. The authors seem to acknowledge this in some responses. For instance, in justifying their superficial application analysis as due to the need to easily summarize their intended point without greater detail.

Our reply: Here we refer to our response given to the point above, which relates to the same issues.

- *Regarding catalyst scale spatial resolution, the point was to establish the relevance, benefits and complimentary nature of the new reactor/instrument with respect to these more established techniques. The point was not to reference SpaciMS specifically. There are many similar, in terms of spatial resolution but individually unique, techniques that have been developed independently (Partridge, Horn, Epling, Goguet, Harold, etc.) across the catalyst community, and these are discussed in the provided references. Again, it is this reviewer’s opinion that the authors shortchange the quality of their manuscript by not orienting their contribution with respect to existing technologies and literature.*

Our reply: We appreciate this clarification because from the first comment given it was not obvious to us that the reviewer was asking for a more thorough *general* literature review of the currently available techniques. As noted, we interpreted it as a request to *specifically* and explicitly also include SpaciMS. The main reason for why we interpreted the original reviewer comment in this way is that we indeed provide references to relevant and very extensive review papers that in detail discuss the large variety of established techniques the reviewer is referring to. Thus, they enable the reader to put our work into perspective in this respect, if more detail is required than what we give in our introduction. To this end, we opted for this solution also largely because this manuscript is under consideration in a journal with a wide scope and very diverse readership. Therefore, we think that a lengthy detailed discussion of the pros and cons of different established techniques would not be suitable for a publication in Nature Communications. Instead, we focus on explaining our technique and

state the spatial and temporal resolution we use, which, together with the provided references to extensive review papers, make it possible for the interested reader to execute a more detailed comparison.

- *It is not meaningless that the behavior of individual catalyst sites differs from the ensemble, but it's not surprising or unknown. What's meaningless is to state his obvious fact without discussing relevance, origin, pathways to advanced performance enabled by the new methodology, etc. It is very nice that this new reactor provides a path for studying variations in individual site nature, and how these influence the bulk performance.*

Our reply: Indeed, we fully agree that individual particle behavior is not surprising as such. However, the ability to directly observe individual particle behavior during catalyst operation in the way we do in the present work is experimentally non-trivial and therefore we believe that the ability to do this is of interest to the catalyst community (as well as the plasmonics and nanofluidics communities). In the manuscript, we explain the individuality by structural and morphological differences between the particles. However, we decided to not be too speculative when discussing the *exact* origin at the level of each individual at this point, since we currently have no direct measure of morphology of the individual particles in question. For the future, we are indeed planning to combine the developed nanoreactor platform and optical readout presented in this manuscript with additional techniques that are capable of identifying more details about the particle morphology, ideally with single-site resolution. To this end, we also would like to highlight that, in our opinion, a work like the present one hopefully will inspire other scientists to join this quest.